# Smooth Multi-Policy Causal Effect Estimation in Longitudinal Settings

Wenxin Chen [1]    Weishen Pan [1]    Kyra Gan [1]    Fei Wang [1]

## Abstract

Comparative evaluation of *multiple* dynamic treatment policies is essential for healthcare and policy decisions, yet conventional longitudinal causal inference methods estimate each in *isolation*, preventing information sharing across counterfactuals. We demonstrate that this separate estimation paradigm induces a structurally uncontrolled second-order bias, inflating finite-sample variance even after standard debiasing with *longitudinal targeted maximum likelihood estimation* (LTMLE). To address this, we propose a policy-aware reparameterization of *Iterative Conditional Expectation* (ICE) Q-functions that enables joint estimation through shared representations. We implement this approach in the **Policy-Encoded Q Network (PEQ-Net)**, an architecture centered on a shared policy encoder. The encoder is trained using kernel mean embeddings, ensuring that the learned representation space reflects population-level policy dissimilarities. After applying an LTMLE correction step, we prove this design imposes a structural constraint on the second-order remainder, thereby stabilizing finite-sample variance. Experiments on semi-synthetic datasets demonstrate that PEQ-Net consistently outperforms existing ICE-based methods, achieving substantial reductions in root-mean-square error, particularly when evaluating closely related policies.

## 1. Introduction

Clinical decision-making often requires comparing the effectiveness of *multiple* treatment strategies from longitudinal data, where decisions adapt to a patient's evolving state (Murphy, 2003; Chakraborty & Moodie, 2013). We are interested in estimating the outcome after multiple treatment decisions (e.g., 30-day mortality) (Xu et al., 2023; Shahn et al., 2020; Nguyen et al., 2010). This task is compli-

cated by the treatment–confounder feedback loop inherent to longitudinal settings: time-varying confounders influence future treatment decisions and are themselves influenced by prior treatments (Robins, 1986; Hernán & Robins, 2010).

Existing methods can be categorized based on their confounding adjustment strategy. *Treatment-model-based* approaches (e.g., IPTW, representation learning (Lim, 2018; Bica et al., 2020; Melnychuk et al., 2022)) model treatment assignment to create a pseudo-population balanced on confounders. However, they are prone to high variance, especially in long horizons, due to extreme propensity scores and the multiplicative accumulation of weights (Farajtabar et al., 2018; Frauen et al., 2025). *Outcome-model-based* approaches avoid modeling treatment assignment altogether. Forward G-computation (Taubman et al., 2009; Li et al., 2021) simulates patient trajectories under a treatment policy, but it requires estimating the full set of covariate and outcome transition models, which can be sensitive to model misspecification in long-horizon and high-dimensional settings. In contrast, *Iterative Conditional Expectation* (ICE) G-computation (Snowden et al., 2011) works backward: it first models the terminal outcome, then recursively steps back in time to model the expected value of future outcomes given the current history, decomposing the problem into a sequence of conditional expectations (Q-functions). By avoiding explicit modeling of the time-varying covariate and reducing reliance on long-horizon simulation, ICE is typically more robust (Bibaut et al., 2019; Wen et al., 2021) and has become a foundation for recent deep learning estimators (Frauen et al., 2023; Shirakawa et al., 2024). However, the standard ICE estimator is a *plug-in* estimator: it is asymptotically inefficient and inconsistent under model misspecification. This limitation leads to the recent development of *doubly robust* (DR) alternatives, e.g., LTMLE (Van der Laan et al., 2011; Díaz et al., 2023).

Despite these advancements, a critical inefficiency remains for *multi-policy evaluation*: whether using a plain ICE estimator or an efficient variant, existing implementations fit a separate sequence of regressions for each candidate policy (Seedat et al., 2022; Bica et al., 2020). They fail to exploit structural similarities across counterfactual trajectories, leading to redundant computation and inflated variance.

We resolve this by innovating on the ICE estimation step.

[1]Cornell University, New York, US. Correspondence to: Fei Wang <few2001@med.cornell.edu>.

*Proceedings of the $43^{rd}$ International Conference on Machine Learning*, Seoul, South Korea. PMLR 306, 2026. Copyright 2026 by the author(s).

We introduce a policy-aware reparameterization of the Q-functions in ICE, enabling information sharing across counterfactual trajectories. To embed the policies, we leverage kernel mean embeddings in a reproducing kernel Hilbert space (RKHS) to define a population-level metric of policy dissimilarity, ensuring the shared representation respects the underlying data-generating process. We apply standard LTMLE debiasing on the resulting ICE estimator. Our contributions are threefold:

- We establish that separately estimated Q-functions incur lack of structural constraint on second-order bias after TMLE debiasing. We introduce a policy-aware reparameterization that preserves the identification of the target parameter while enabling shared learning (Section 4.1).
- We propose the **Policy-Encoded Q Network (PEQ-Net)**, a theoretically grounded architecture featuring a shared policy encoder that reflects maximum mean discrepancy. After applying LTMLE debiasing, we prove this design imposes a structural constraint on the aforementioned bias of the treatment effect estimator, stabilizing the finite-sample variance (Theorem 4.2).
- We demonstrate the effectiveness of our method through extensive experiments on semi-synthetic datasets derived from MIMIC, together with a real-world case study of vasopressor treatment in sepsis patients. On the semi-synthetic data, our method substantially reduces root mean square error, particularly when evaluating closely related policies. Code is available at `https://github.com/Wenxin-Elmon-Chen/PEQ`.

### 1.1. Related Work

**Longitudinal Treatment Effect Estimation** Classic *treatment-model–based methods* use *Inverse Probability of Treatment Weighting* (IPTW) (Rosenbaum & Rubin, 1983; 1984; Robins, 2000; Hernán & Robins, 2006; Cole & Hernán, 2008), its stabilized variant (Xu et al., 2010; Chesnaye et al., 2022), or matching (Austin, 2011; Stuart, 2010) to create balanced pseudo-populations. Recent deep learning methods adopted this principle and learn latent representations that predict outcomes while achieving a balance between treatment arms (Lim, 2018; Bica et al., 2020; Seedat et al., 2022; Melnychuk et al., 2022). These algorithms often perform poorly with finite samples in long horizons due to the compounding of small model errors in the treatment models, leading to high variance or residual bias (Bica et al., 2020; Petersen et al., 2012; Cole & Hernán, 2008).

*Outcome-model–based methods* directly model the outcome-generating process to estimate treatment effects using *G-computation* (Robins, 1986; Snowden et al., 2011) by standardizing over the natural course of confounder distributions. Popular forward G-computation includes Chiu et al. (2023); McGrath et al. (2020); Li et al. (2021); Rein et al. (2024).

Recently, ICE G-computation gained attention due to improved stability from its recursive formulation that mitigates error propagation in time-varying covariates across time steps (Oprescu et al., 2025; Liu et al., 2025; Frauen et al., 2025; Shirakawa et al., 2024; Frauen et al., 2023).

*Doubly robust variants*, combining treatment and outcome models, enhance robustness against model misspecification, improving finite-sample performance in high-dimensional, long-horizon settings. These include LTMLE (Van der Laan et al., 2011; Stitelman et al., 2012; Rosenblum & Van Der Laan, 2010; Díaz et al., 2023) and recent deep learning variants (Shirakawa et al., 2024; Frauen et al., 2023; Guo et al., 2024; Frauen et al., 2025; Hess et al., 2024).

**Off-Policy Evaluation** Our problem relates to off-policy evaluation (OPE) in reinforcement learning, which estimates a target policy's mean reward using data from a different behavior policy. Methodologically, OPE methods mirror the standard categories in longitudinal causal estimation and face the same core challenges: 1) *importance sampling* (related to IPTW) reweights trajectories induced by the behavior policy with that of the target policy (Mahmood et al., 2014; Thomas & Brunskill, 2016b; Hanna et al., 2019; Schlegel et al., 2019), but often suffer from high variance over long horizons (Liu et al., 2018). Recent works exploit the Markov assumption to reduce variance (Bossens & Thomas, 2024; Xie et al., 2019; Shen et al., 2021; Fujimoto et al., 2021), but this is often violated in our setting where outcomes can depend on full history. 2) *Direct method* (analogous to G-computation) model system dynamics to simulate counterfactual trajectories (Le et al., 2019; Duan et al., 2020; Voloshin et al.). *DR methods* improve finite-sample efficiency and retain consistency when either component is correctly specified (Bibaut et al., 2019; Jiang & Li, 2016; Thomas & Brunskill, 2016a; Kallus & Uehara, 2020; Dudík et al., 2011).

Our work diverges from prior methods of longitudinal treatment effect estimation or OPE in RL by targeting the problem of estimation stability under the joint evaluation of multiple treatment policies. To achieve this, we introduce a modification to the standard Q-function (outcome model) training procedure and subsequently apply an LTMLE step to correct for the plug-in bias of the initial estimator.

## 2. Problem Setup and Preliminary

We observe $n$ i.i.d. longitudinal trajectories, each of length $\tau$, generated under a *unknown behavior policy*, $\boldsymbol{\pi}^*$. For each trajectory and at each discrete time point $t \in \{1, \cdots, \tau\}$, we observe time-varying covariates $L_t$, followed by a binary treatment assignment $A_t \in \mathcal{A} = \{0, 1\}$. A final outcome $Y$ is observed after time $\tau$. Thus, one trajectory observed for a single unit is $O = (L_1, A_1, L_2, A_2, \ldots, L_\tau, A_\tau, Y) \sim P_0$, where $P_0$ is the unknown true distribution induced by the

*unknown behavior policy* that generated the data. (Any intermediate outcome observed after $A_t$ can be absorbed into $L_{t+1}$ without loss of generality.)

We use subscripts for temporal sequences, e.g., $X_{t:\tau} = (X_t, X_{t+1}, \ldots, X_\tau)$. The history available before assigning treatment $A_t$ is $H_t = \{L_i, A_i\}_{i=1}^{t-1} \cup L_t$, taking values in a history space $\mathcal{H}_t$. Both covariates and treatments may depend on the entire preceding history: $L_t$ may be influenced by $H_{t-1} \cup \{A_{t-1}\}$, while $A_t$ is determined by $H_t$.

Treatments in the observed data are assigned according to the unknown behavior policy $\boldsymbol{\pi}^* = (\pi_1^*, \cdots, \pi_\tau^*)$. Formally, each $\pi_t^* : \mathcal{H}_t \to (0, 1)$ specifies the probability of receiving treatment given the history, i.e., $\pi_t^*(H_t) = P(A_t = 1 | H_t)$.

**Problem Setup** We consider a set of $K$ dynamic counterfactual treatment policies, $\{\boldsymbol{\pi}^{(1)}, \cdots, \boldsymbol{\pi}^{(K)}\}$, that we wish to evaluate and compare. For each policy $\boldsymbol{\pi}^{(i)}$, let

$$\dot{a}_{1:\tau}^{(i)} = (\pi_1^{(i)}(H_1), \ldots, \pi_\tau^{(i)}(H_\tau)) \qquad (1)$$

denote the sequence of counterfactual actions it would assign given the observed history, and let $Y(\boldsymbol{\pi}^{(i)})$ be the corresponding potential outcome. The **conditional average potential outcome (CAPO)** for policy $\boldsymbol{\pi}^{(i)}$ given baseline covariates $L_1$ is defined as $\psi^{(i)} = \mathbb{E}[Y(\boldsymbol{\pi}^{(i)}) | L_1]$.

While each CAPO measures the expected outcome under a single policy, the ultimate quantity that informs a decision between two policies $\boldsymbol{\pi}^{(i)}$ and $\boldsymbol{\pi}^{(j)}$ is the difference between their CAPOs. Following the convention (Frauen et al., 2023; Oprescu et al., 2025), we call this difference in CAPOs the **conditional average treatment effect** (CATE): $\mathbb{E}[Y(\boldsymbol{\pi}^{(i)}) | L_1] - \mathbb{E}[Y(\boldsymbol{\pi}^{(j)}) | L_1]$.

We use $\boldsymbol{\pi}$ to denote a generic *counterfactual* policy. Throughout, we assume the standard longitudinal causal inference conditions to ensure the identifiability of our target parameters (Shirakawa et al., 2024): (1) Consistency: $Y(\boldsymbol{\pi}) = Y$ when $\boldsymbol{\pi}^* = \boldsymbol{\pi}$; (2) Sequential Ignorability: $Y(\boldsymbol{\pi}) \perp A_t \mid H_t \;\; \forall t \in \{1, \ldots, \tau\}.$; (3) Positivity: $\forall t \in \{1, \ldots, \tau\}, \boldsymbol{\pi}(H_t)/\boldsymbol{\pi}^*(H_t) < \infty$.

The rest of the section reviews two key methodological building blocks. We first describe ICE G-computation (Sec. 2.1), which constructs plug-in estimates of dynamic treatment effects. We then review LTMLE (Sec. 2.2), a debiasing procedure that corrects the first-order plug-in bias of initial estimators, including those from ICE.

## 2.1. ICE G-computation

ICE G-computation estimates the CAPO through backward recursion based on the G-computation formula (Robins & Hernan, 2008; Bang & Robins, 2005), which expresses the CAPO as a nested sequence of conditional expectations

evaluated under the counterfactual policy $\boldsymbol{\pi}$:

$$\psi = \mathbb{E}_{\dot{a}_1 \sim \pi_1(H_1)}\Big[\mathbb{E}_{\dot{a}_2 \sim \pi_2(H_2)}\Big[\ldots$$
$$\mathbb{E}_{\dot{a}_\tau \sim \pi_\tau(H_\tau)}[Y | A_\tau = \dot{a}_\tau, H_\tau] \ldots | A_1 = \dot{a}_1, L_1]\Big]\Big]. \quad (2)$$

ICE estimates this quantity by breaking down the nested expectation into a series of recursive regression problems, proceeding backward from $t = \tau$ to $t = 1$. Define $Q_{\tau+1}^{\text{ICE}} = Y$. For each step $t = \tau, \ldots, 1$, ICE recursively defines

$$Q_t^{\text{ICE}}(A_t, H_t) = \mathbb{E}[Q_{t+1}^{\text{ICE}}(\pi_t(H_{t+1}), H_{t+1}) | A_t, H_t]. \quad (3)$$

The CAPO is then identified as $\psi = \mathbb{E}[Q_1^{\text{ICE}}(\pi_1(H_1), H_1)]$.

In practice, ICE is implemented by iterating the following two steps from $t = \tau$ to 1:

1. Fit a regression model $Q_t^{\text{ICE}}(A_t, H_t)$ to predict the pseudo-outcome $\hat{Q}_{t+1}^{\text{ICE}}(\pi_{t+1}(h_{t+1}), h_{t+1})$. At $t = \tau$, the pseudo-outcome is $\hat{Q}_{\tau+1}^{\text{ICE}} = Y$ by construction.
2. Generate the pseudo-outcome for the previous step $t - 1$ by imputing the treatment according to the counterfactual policy, yielding $\hat{Q}_t^{\text{ICE}}(\pi_t(h_t), h_t)$.

The final estimate of $\psi$ is obtained by averaging $\hat{Q}_1^{\text{ICE}}(\pi_1(h_1), h_1)$ over the sample.

*Remark* 2.1. We note that the pseudo-outcome at each step is conditioned on the observed history $h_{t+1}$, not the history altered by $\boldsymbol{\pi}$. While this may appear counterintuitive, the pseudo-outcome represents the conditional mean outcome given current history, under the intervention that follows $\boldsymbol{\pi}$ from $t + 1$ onward. Under the standard assumptions, this conditional expectation is identifiable from the observed data. Further, while this vanilla formulation fits separate $Q_t^{\text{ICE}}$ at each time step, modern approaches often employ shared autoregressive architectures, such as LSTM or Transformer, to learn a single time-aware $Q$-function across all time steps. This implementation stabilizes the entire training process and achieves better finite-sample performance.

## 2.2. LTMLE

While ICE G-computation provides a plug-in estimator that is consistent under correct nuisance specification, its theoretical guarantees become difficult to maintain when using highly flexible function approximators such as transformers, which may overfit or introduce regularization bias. To safeguard against such inconsistencies, we adopt LTMLE as a subsequent debiasing step.

We consider a nonparametric model class for the nuisance functions (Lin et al., 2026; Li & Gan, 2025). LTMLE corrects first-order bias by updating the initial nuisance estimates so that the influence function (IF) of the target parameter is set to zero in the sample. This yields an estimator

that is consistent even when one of the nuisance models (outcome or treatment) is misspecified, and it improves the estimator's asymptotic normality and efficiency.

Concretely, let the target parameter $\psi = \Psi(P)$ be a functional of the data-generating distribution $P$. In practice, $\Psi(P)$ depends on nuisance components–in our case, the propensity score and outcome model at each time step. A plug-in estimator $\Psi(\hat{P})$ is obtained by substituting estimates of these nuisances. The error of this estimator can be expanded via a von Mises expansion (Van der Laan et al., 2011; Cho et al., 2024):

$$\hat{\psi}_n - \psi_0 = \mathbb{P}_n D_\Psi^*(P_0) - \underbrace{\mathbb{P}_n D_\Psi^*(\hat{P}_n)}_{\text{plug-in bias}} +$$

$$\underbrace{(\mathbb{P}_n - P_0)[D_\Psi^*(\hat{P}_n) - D_\Psi^*(P_0)]}_{\text{empirical process term}} + \underbrace{Rem(\hat{P}_n, P_0)}_{\text{second-order remainder}} , \quad (4)$$

where $D_\Psi^*(P)$ is the IF for the target parameter $\psi$, and $\mathbb{P}_n$, $P_0$ the empirical and true expectation operators, respectively; $Rem(\hat{P}_n, P_0)$ is the second-order remainder in the difference between $\hat{P}_n$ and $P_0$. At a high level, LTMLE achieves asymptotic efficiency by eliminating the *plug-in bias term* through an iterative procedure, while relying on the empirical process and the second-order remainder terms to be $o_p(n^{-1/2})$. See prior work (Van der Laan et al., 2011; Cho et al., 2024; Li & Gan, 2025; Shirakawa et al., 2024) for a detailed description of this procedure.

The *empirical process term* is typically controlled through regularity conditions on the nuisance estimators, such as Donsker-type assumptions or sample-splitting procedures (Kosorok, 2008; Van der Laan et al., 2011; Chernozhukov et al., 2018; Newey & Robins, 2018)

The second-order remainder, which captures higher-order interactions between nuisance estimation errors (such as those of the outcome regression and treatment mechanism) must satisfy $\sum_{t=1}^{\tau} \|\hat{\pi}_t - \pi_t^*\| \|\hat{Q}_t - Q_t^*\| = o_p(n^{-1/2})$. This condition becomes substantially more demanding as $\tau$ grows, because ICE-based Q-estimation propagates errors backward through the recursion. When constructing a CATE estimator as the difference of two CAPO estimates, the remainder bias compounds further, making stable control of this term critical for finite-sample performance. We formalize this statement in Sec. 3.

Our theoretical analysis therefore focuses on the second-order remainder, examining how refinements to ICE G-computation can achieve better-controlled remainder terms and improved robustness. A central challenge arises when estimating CAPO contrasts across multiple treatment policies: to ensure sufficient estimation accuracy for each policy, we must explicitly incorporate a notion of policy "similarity" into the estimation process itself. This allows information to be shared across policies while preserving

modeling flexibility. We formalize this approach and its benefits for remainder control in Section 4.

## 3. Limitation of Current Paradigm

From the recursive definition in Eq. (3), note that $Q_t^{\text{ICE}}$ depends on the policy only through its future tail $\pi_{t+1:\tau}$ because it is defined in terms of $Q_{t+1}^{\text{ICE}}$, which in turn depends on $\pi_{t+2:\tau}$. Consequently, *two policies that coincide from time $t + 1$ onward should induce the same $Q_t^{\text{ICE}}$ function.*

However, existing frameworks generally do not follow this behavior. To estimate the CATE between two policies, these methods (e.g., DeepACE, DeepLTMLE) adopt a separate estimation paradigm: For two policies $\boldsymbol{\pi}^{(i)}$ and $\boldsymbol{\pi}^{(i)}$, two full sets of nuisance models are trained independently, which include the outcome and propensity score models at all time steps, and then CATE is obtained by differencing the resulting CAPO estimates. Consequently, the models minimize two different objectives, making the $Q$-functions implicitly depend on the entire policy trajectories, not only the future tail. (The propensity score model $\hat{\pi}$ also implicitly depends on the counterfactual because of the multi-task learning paradigm.) Such a decoupling of nuisance estimation across policies induces unnecessary error in the estimation of $Q$ (and $\hat{\pi}$), inflating bias in CATE estimation.

The following lemma (proof in Appendix B.1) formalizes the statistical consequence of this decoupling by an extreme condition: Even when two counterfactual policies behave identically, the contrast of their second-order remainders, $\text{Rem}^{(i),(j)} = \text{Rem}^{(i)} - \text{Rem}^{(j)}$, generally does not cancel. This lack of structural cancellation implies that CATE may exhibit irregular behavior driven by nuisance estimation, rather than by meaningful policy differences.

**Lemma 3.1.** *Consider the separate estimation paradigm where nuisance models are subject to independent random initialization. Even when two policies behave exactly the same at all steps, the second-order remainder in CATE generally does not cancel, i.e., $\text{Rem}^{(i),(j)} \neq 0$ almost surely.*

## 4. Methods

This section introduces our method. We first leverage a key structural property of ICE G-computation to reparameterize $Q$-function such that it enables model sharing across policies (Section 4.1). We then formalize policy distance on RKHS and construct policy representations that preserve these distances (Section 4.2). Finally, we combine these components and propose our model architecture (Section 4.3).

### 4.1. Reparametrization of Q-Function

Existing methods independently fit separate outcome regressions for each counterfactual policy, which violates a crucial

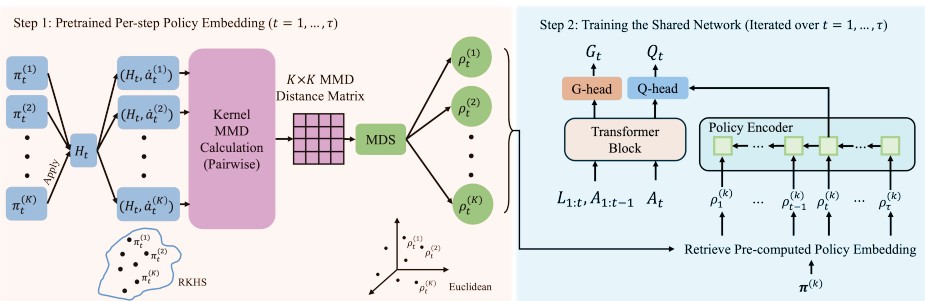

*Figure 1.* Illustration of the PEQ-Net. Step 1 computes per-step policy embeddings using pairwise MMD distances followed by MDS. Step 2 aggregates the resulting embeddings with a policy encoder and conditions the shared $Q$-functions on the encoded policy representation.

requirement for smooth policy contrasts. To address this, we propose to explicitly parameterize the outcome regression by the future policy tail, shifting this dependency from an implicit to an explicit representation.

Specifically, we redefine the $Q$-function at time $t$ to condition not only on the current action and history, but also on the remaining policy sequence $\pi_{t+1:\tau}$:

$$Q_t(A_t, H_t, \pi_{t+1:\tau}) = \mathbb{E}[Q_{t+1}(\pi_{t+1}(H_{t+1}), H_{t+1}, \pi_{t+2:\tau})|A_t, H_t, \pi_{t+1:\tau}]. \quad (5)$$

This formulation allows a single model to represent $Q_t$ across multiple counterfactual policies, because the policy tail $\pi_{t+1:\tau}$ is now an explicit input.

We show that this reparameterization preserves identification of the target estimand (Proof in Appendix B.2).

**Lemma 4.1** (Identification of reparameterized $Q$-functions). *For any policy $\boldsymbol{\pi}^{(k)}$, the reparameterized outcome regressions $\{Q_t(\cdot, \cdot, \pi_{t+1:\tau}^{(k)})\}_{t=1}^{\tau}$ defined in Eq. (5) identify the same CAPO as standard ICE G-computation. In particular,*

$$\mathbb{E}\left[Q_1^{\text{ICE}}(\pi_1^{(k)}(H_1), H_1)\right] = \mathbb{E}\left[Q_1(\pi_1^{(k)}(H_1), H_1, \pi_{2:\tau}^{(k)})\right].$$

### 4.2. RKHS-Based Policy Embedding

To enable joint estimation, we represent each policy (function) by a fixed-dimensional vector that captures its behavior relative to other policies. The core idea is to embed policies into a *continuous metric space* where distances reflect the discrepancy between the distributions of actions they would generate given the observed patient histories. This geometric structure allows the estimation model to share statistical strength across policies that behave similarly, thereby stabilizing the learning process.

Formally, we construct this embedding using the framework of reproducing kernel Hilbert spaces (RKHS). Let $h_t^{\{\ell\}}$ denote the observed history at time $t$ for individual $\ell$. At each time step $t$, for a policy $\pi_t$, we apply it to all *observed*

*covariates* $\{h_t^{\{\ell\}}\}_{\ell \in [n]}$ to generate a corresponding set of actions. Let $z_t^{\{\ell\}}$ be the history action pair that is induced by policy $\pi_t$ on individual $\ell$, i.e., $z_t^{\{\ell\}} = \left(h_t^{\{\ell\}}, \pi_t(h_t^{\{\ell\}})\right)$. The pairs $\left(z_t^{\{\ell\}}\right)_{\ell \in [n]}$ form an empirical distribution over the space $\mathcal{Z}_t = \mathcal{H}_t \times \mathcal{A}$ that characterizes how the policy behaves on the available data. To quantify similarity between two policies, we measure the distance between their induced empirical distributions using the **maximum mean discrepancy** (MMD) (Gretton et al., 2012a; Sejdinovic et al., 2013; Chwialkowski et al., 2015; Gretton et al., 2012b). This kernel-based distance equips the space of policies with a metric that respects distributional shape, providing a theoretically well-founded notion of policy dissimilarity.

Since the history $h_t$ grows in dimension with $t$, we define a *time-specific characteristic kernel* $\kappa_t : \mathcal{Z}_t \times \mathcal{Z}_t \to \mathbb{R}$ for each step (we use Gaussian kernels with bandwidths adapted to the dimension of $\mathcal{Z}_t$). Let $\mathcal{F}_t$ be the RKHS induced by $\kappa_t$. The mean embedding of the policy $\pi_t^{(i)}$, $\mu_t^{(i)}$, is defined as

$$\mu_t^{(i)} = \frac{1}{n} \sum_{\ell=1}^{n} \kappa_t\left(\,\cdot\,, (z_t)\right) \in \mathcal{F}_t.$$

---

**Algorithm 1** RKHS-Based Policy Embedding

---

1: **Input**: Dataset $\{O_i\}_{i=1}^n$, policies $\{\boldsymbol{\pi}^{(k)}\}_{k=1}^K$
2: **Output**: Policy embeddings $\{\{\rho_t^{(k)}\}_{k=1}^K\}_{t=1}^{\tau}$
3: **for** $t = 1$ to $\tau$ **do**
4:    Initialize $K \times K$ distance matrix $D_t$
5:    **for** $k = 1$ to $K$ **do**
6:       Apply $\pi_t^{(k)}$ to $H_t$ to obtain $Z_t^{(k)} = (H_t, \dot{a}_t^{(k)})$
7:    **end for**
8:    **for** $i, j \in [K]$ **do**
9:       $D_t[i, j] \leftarrow \|\mu_t^{(i)} - \mu_t^{(j)}\|_{\mathcal{F}_t}$
10:    **end for**
11:    Normalize $D_t$ by $\max(D_t)$
12:    Apply metric MDS to $D_t$ to obtain $\{\rho_t^{(k)}\}_{k=1}^K$ s.t. $\|\rho_t^{(i)} - \rho_t^{(j)}\| \approx \|\mu_t^{(i)} - \mu_t^{(j)}\|$
13: **end for**

---

The squared MMD between policies $\pi_t^{(i)}$ and $\pi_t^{(j)}$ is then the squared RKHS distance between their embeddings

$$\mathrm{MMD}^2(\pi_t^{(i)}, \pi_t^{(j)}) = \|\mu_t^{(i)} - \mu_t^{(j)}\|_{\mathcal{F}_t}.$$

Given a set of $K$ counterfactual policies, we compute, for each time $t$, all pairwise MMD distances using $\kappa_t$, resulting in a symmetric distance matrix $D_t \in \mathbb{R}^{K \times K}$ with entries $D_t[i,j] = \mathrm{MMD}^2(\pi_t^{(i)}, \pi_t^{(j)})$.

To obtain a fixed-dimensional vector representation suitable for neural-network input, we apply **metric multidimensional scaling** (MDS) (Kruskal, 1964) to each matrix $D_t$. Let $d$ be the dimension of the low-dimensional representation, then MDS finds $\rho_t^{(i)} \in \mathbb{R}^d$ that preserve the original MMD geometry as closely as possible:

$$\|\rho_t^{(i)} - \rho_t^{(j)}\|_2 \approx D_t[i,j].$$

Formally, MDS solves the following optimization problem

$$\min_{\rho_t^{(1)},\ldots,\rho_t^{(K)} \in \mathbb{R}^d} \sum_{i<j} \big(D_t[i,j] - \|\rho_t^{(i)} - \rho_t^{(j)}\|_2\big)^2. \quad (6)$$

This optimization problem is solved by the SMACOF algorithms (Borg & Groenen, 2005).

These precomputed policy embeddings $\{\rho_t^{(i)}\}$ serve as structured, continuous representations of each policy's behavioral profile (Appendix C.7 provides empirical analyses of the embeddings). By supplying them as inputs to our architecture (Sec. 4.3), the model can explicitly leverage the metric relationships among policies, enabling stable joint estimation through shared representation learning.

### 4.3. Our Architecture: PEQ-Net

Building on the policy embeddings described above, we now detail how these structured representations enable joint learning across policies. Algorithm 2 summarizes the training procedure of PEQ-Net. The core design trains a **single shared set of outcome and propensity models** across all $K$ counterfactual policies, and the outcome regression is explicitly conditioned on the **policy-tail embedding** $\tilde{\rho}(\rho_{t+1}^{(i)})$.

**Architecture** We adopt the decoder-only Transformer architecture from DeepLTMLE as the backbone (Shirakawa et al., 2024). Inputs are first passed through a type-embedding layer distinguishing covariates $L$, treatment $A$, and outcome $Y$, and combined with positional encodings. The resulting sequence is processed by a Transformer block $\phi(H_t, A_t)$, whose output feeds into a $G$-head for propensity estimation and a $Q$-head for outcome regression.

Although our notation distinguishes the time-indexed models $\{Q_t, G_t\}_{t=1}^{\tau}$, we implement these functions by the same $Q$-head and $G$-head with shared parameters. At step $t$, $G$

---

**Algorithm 2** PEQ-Net Training

1: **Input:** Training Data $\{O_i\}_{i=1}^n$; counterfactual policies $\{\boldsymbol{\pi}^{(k)}\}_{k=1}^K$ with corresponding treatments $\{\dot{a}_{1:\tau}^{(k)}\}_{k=1}^K$ defined in Eq (1); embedding $\{\{\rho_t^{(k)}\}_{k=1}^K\}_{t=1}^{\tau}$; target network coefficient $\beta$.

2: **Output:** parameter $\theta$

3: Initialize model blocks $q, g, \phi, \tilde{\rho}$, with parameters $\theta$

4: **for** batch=1,…,B **do**

5:    $Q_{\tau+1} = Y$

6:    **for** $t = 1, \ldots, \tau$ **do**

7:       **for** $k = 1, \ldots, K$ **do**

8:          G-comp target generation using target network $\theta'$:  $\hat{Q}_{t+1}^{(k)}(\dot{a}_{t+1}^{(k)}, H_{t+1}) = q_{\theta'}(\phi_{\theta'}(H_{t+1}, \dot{a}_{t+1}^{(k)}), \tilde{\rho}_{\theta'}(\rho_{t+2:\tau}^{(k)}))$

9:          G-comp prediction using current network $\theta$:  $Q_t^{(k)}(A_t, H_t) = q_\theta(\phi_\theta(H_t, A_t); \tilde{\rho}_\theta(\rho_{t+1:\tau}^{(k)}))$  $\mathcal{L}_t^{Q,(k)} = \mathcal{L}_{mse}(Q_t^{(k)}(A_t, H_t), \hat{Q}_{t+1}^{(k)}(\dot{a}_{t+1}^{(k)}, H_{t+1}))$

10:       **end for**

11:       $G(H_t) = g(\phi(H_t))$

12:       $\mathcal{L}_t^G = \mathcal{L}_{bce}(G(H_t), A_t)$

13:    **end for**

14:    $\mathcal{L}(\theta) = \sum_{t=1}^{\tau} \left[ \mathcal{L}_t^G + \sum_{k=1}^K \mathcal{L}_t^{Q,(k)} \right]$

15:    $\theta \leftarrow \theta - \eta \nabla_\theta L$

16:    $\theta' \leftarrow \beta \cdot \theta + (1 - \beta) \cdot \theta'$

17: **end for**

---

has access to $H_t$, while $Q$ has access to $(H_t, A_t)$; this is enforced by masking future variables in the input sequence.

To condition $Q$ on future policies, we introduce a policy-tail encoder $\tilde{\rho}$ that aggregates the per-time-step policy embeddings $\rho_{t+1:\tau}$. We implement $\tilde{\rho}$ as an RNN operating backward in time, mapping $\rho_{t+1:\tau}$ to a fixed-dimensional embedding $\tilde{\rho}(\rho_{t+1:\tau}) \in \mathbb{R}^p$, which is concatenated with the Transformer block representation and passed to the $Q$-head, yielding the reparameterized outcome model

$$Q_t = q\big(\phi(H_t, A_t), \tilde{\rho}(\rho_{t+1:\tau})\big).$$

**Training** Training follows Algorithm 2. At each time step $t$ and for each counterfactual policy $k$, we first generate ICE G-computation targets by evaluating the shared outcome model at the next time step under the counterfactual action $\dot{a}_{t+1}^{(k)}$ (Lines 8–9). The outcome model is then updated by minimizing a mean squared error loss between the current prediction and the generated target (Lines 10–12).

To stabilize training, we introduce a lagged target network. Specifically, we maintain a slowly updated copy of the parameters $\theta'$ and use it to compute the regression target $\hat{Q}_{t+1}$, while updating $\theta$ via gradient descent. The target network is updated using Polyak averaging ($\theta' \leftarrow \beta \theta + (1 - \beta) \theta'$) after each batch, which reduces the moving-target issue by

decoupling target generation from parameter updates. We set $\beta = 0.005$ throughout.

The treatment model is trained in parallel using a binary cross-entropy loss on observed actions (Lines 14–15). All losses are summed across time steps and counterfactual policies to form the final training objective.

**Inference** After training, we evaluate each counterfactual policy independently using a one-step LTMLE update on a held-out evaluation set. The learned outcome and treatment models provide initial nuisance estimates, which are then targeted using regularized longitudinal TMLE (Bibaut et al., 2019; Shirakawa et al., 2024) (Appendices A.1 and A.2).

**Guarantee** The next theorem analyzes how the second-order remainder behaves as policies vary (Proof in Appendix B.6). To formalize policy similarity, we measure policy distance at the trajectory level using the MMD between policy-induced trajectory distributions, defined on a product RKHS $\mathcal{F}_{1:\tau}$ (Appendix B.3).

**Theorem 4.2** (Lipschitz control of the CATE second-order remainder). *Suppose the parameterized outcome model and the policy encoder are Lipschitz in their respective inputs, and that the true outcome model and the policy-induced density product are Lipschitz continuous with respect to policy-induced trajectory distributions (detailed in Assumption B.2). Then there exists a finite constant $L_R < \infty$ such that the magnitude of the second-order remainder in the CATE policy contrast is controlled by the trajectory-level policy distance:* $\left|\mathrm{Rem}^{(i),(j)}\right| \le L_R \|\mu_{1:\tau}^{(i)} - \mu_{1:\tau}^{(j)}\|_{\mathcal{F}_{1:\tau}}$.

This control of the remainder mitigates spurious non-smooth behavior that arise in the separate estimation paradigm. The Lipschitz condition on the parametrized outcome model and policy encoder are standard in machine learning. The smooth condition of true outcome models is mild, ensuring that counterfactual outcomes vary smoothly for small changes in the policy. The regularity condition on policy-induced density product ensures that it varies smoothly with the policy, preventing small policy changes from being amplified multiplicatively over time. We empirically validate this control through numerical experiments.

## 5. Experiments

We evaluate PEQ-Net on semi-synthetic datasets derived from MIMIC-III (Johnson et al., 2016), a real-world clinical dataset. Section 5.1 outlines the experimental setup. Section 5.2 demonstrates that PEQ-Net consistently reduces bias and variance, and Section 5.3 presents ablation studies.

### 5.1. Setup

**Baselines** We compare PEQ-Net against longitudinal CATE estimators based on ICE G-computation. Specifi-

cally, we include (1) ICE G-computation with Super Learner, (2) LTMLE with Super Learner, (3) DeepACE, and (4) DeepLTMLE, which represent increasing levels of modeling flexibility within this paradigm. See Appendix C.3 for implementation details.

**Datasets** We first adopt the semi-synthetic data-generating process (DGP) introduced in DeepACE, which includes 10 time-varying real-world measurements in MIMIC-III, along with synthesized treatment assignments and outcomes. In this setting, treatment does not affect time-varying covariates, but only the intermediate outcomes, which induces a limited treatment-confoundeder feedback loop. We then extend this DGP by introducing five additional synthesized time-varying covariates that both influence and are influenced by treatment, resulting in a expanded treatment–confounder feedback structure. Detailed descriptions of both DGPs are provided in Appendix C.1.

For each DGP, we sample $N = 1000$ patient trajectories from the MIMIC-III dataset, and then synthesize data based on the real-world observed covariates over $\tau = 15$ time steps. We use 800 trajectories for training and 200 for validation to select optimal hyperparameters (See Appendix C.4). The selected hyperparameters are then used to retrain the nuisance models on the combined set of 1000 trajectories. Final CATE estimates are obtained by applying the retrained nuisance models to the same 1000 trajectories.

**Counterfactual Policy** We consider two classes of counterfactuals:

(a) *deterministic treatment sequences*, which are standard for benchmarking in the longitudinal CATE literature and supported by all baseline methods.

(b) *dynamic treatment policies*, which map observed histories to treatment decisions.

Most baselines are designed for (a), while PEQ-Net accommodates both (a) and (b). Accordingly, we evaluate the two classes separately.

For (a), we follow the setup of DeepACE and evaluate all baselines on three counterfactual settings of this class. In this case, we bypass the MDS step and directly input the deterministic sequence into the policy encoder, treating it as a degenerate policy embedding.

For (b), we restrict comparisons to DeepLTMLE, which is designed to handle dynamic policies without architectural changes. We consider two scenarios with different levels of policy similarity: (1) policies differ only in the early steps and coincide thereafter, creating substantial structural overlap; and (2) policies differ at every step, eliminating trivial overlap and serving as a stress test for policy-aware representation learning. Detailed counterfactual setups are provided in Appendix C.2.

*Table 1.* Results on semi-synthetic DGPs with counterfactuals of deterministic sequences.

| Model | Limited Time-varying Confounding | | | | | | Expanded Time-varying Confounding | | | | | |
|---|---|---|---|---|---|---|---|---|---|---|---|---|
| | Bias | | | RMSE | | | Bias | | | RMSE | | |
| | CF 1a | CF 2a | CF 3a | CF 1a | CF 2a | CF 3a | CF 1a | CF 2a | CF 3a | CF 1a | CF 2a | CF 3a |
| G-comp (sl.) | 1.62±1.19 | 1.31±1.06 | 1.73± 0.90 | 1.99 | 1.67 | 1.94 | 1.64± 0.90 | 1.04 ± 1.26 | 1.58 ± 0.83 | 1.86 | 1.60 | 1.78 |
| LTMLE (sl.) | 2.92±1.54 | 3.66±1.95 | 1.92±1.61 | 3.28 | 4.12 | 2.47 | 1.89± 1.51 | 2.41± 2.06 | 2.41±1.47 | 2.40 | 3.13 | 2.80 |
| DeepACE | 2.15±0.51 | 0.51±0.47 | 1.98±0.77 | 2.21 | 0.69 | 2.12 | 2.42±0.45 | 0.49±0.26 | 2.21±0.52 | 2.46 | 0.55 | 2.26 |
| D.LTMLE | 0.77±0.61 | 0.71±0.61 | 0.93±0.97 | 0.97 | 0.92 | 1.33 | 0.81±0.75 | 1.13±0.88 | 0.58±0.58 | 1.09 | 1.42 | 0.81 |
| PEQ-Net | **0.53±0.38** | **0.26±0.22** | **0.21±0.18** | **0.65** | **0.34** | **0.27** | **0.63±0.37** | **0.20±0.15** | **0.26±0.22** | **0.72** | **0.25** | **0.34** |

*Table 2.* Results on counterfactuals of dynamic treatment policy.

| DGP | Model | Policies differ only in the first two steps | | | | Policies differ in all steps | | | |
|---|---|---|---|---|---|---|---|---|---|
| | | Bias | | RMSE | | Bias | | RMSE | |
| | | CF 1b | CF 2b | CF 1b | CF 2b | CF 1c | CF 2c | CF 1c | CF 2c |
| Limited | D.LTMLE | 0.0998±0.0780 | 0.1006±0.1068 | 0.1255 | 0.1448 | 0.2276± 0.1538 | 0.2310 ± 0.1357 | 0.2725 | 0.2662 |
| | PEQ-Net | **0.0015±0.0021** | **0.0132±0.0070** | **0.0026** | **0.0148** | **0.0969 ± 0.0761** | **0.1421 ± 0.0757** | **0.1220** | **0.1601** |
| Expanded | D.LTMLE | 0.1024±0.0825 | 0.1386±0.1208 | 0.1302 | 0.1818 | 0.2986± 0.2227 | 0.4320±0.2492 | 0.3692 | 0.4956 |
| | PEQ-Net | **0.0003±0.0001** | **0.0030±0.0017** | **0.0003** | **0.0034** | **0.1130 ± 0.0883** | **0.2530 ± 0.1012** | **0.1420** | **0.2715** |

For each method and counterfactual setting, we repeat 20 experiments on data generated with different random seeds, and then report the absolute bias and root mean squared error (RMSE) between the estimated and true CATE.

## 5.2. Results

Tables 1 and 2 summarize performance across DGPs and counterfactual types. PEQ-Net consistently achieves the lowest bias and RMSE. Despite sharing the same backbone with DeepLTMLE, PEQ-Net yields substantial error reductions across counterfactuals, highlighting the benefit of policy-aware model sharing.

## 5.3. Ablation Study

**Improvement in ICE G-computation** ($Q$) Our final estimator uses LTMLE, which depends on both the outcome regression ($Q$) and the treatment model ($G$). To isolate the contribution of the proposed policy encoder to the learned $Q$-functions, we remove the LTMLE targeting update for both DeepLTMLE and PEQ-Net and report CATE estimates directly from the fitted $Q$. Tables C.2 and C.3 in Appendix C.5 show that PEQ-Net continues to outperform DeepLTMLE under this ICE-only evaluation, indicating that the primary improvement is attributable to the outcome regression $Q$, rather than to reduced variance from the treatment model $G$.

**Alternative multi-policy evaluation strategy** Beyond fully separate estimation, two natural parameter-sharing alternatives for multi-policy evaluation are: (1) learning a shared backbone and subsequently fine-tuning it independently for each target policy, and (2) learning a model with a shared backbone and multiple Q-heads, where each policy is associated with a dedicated Q-head. We evaluate these alternatives using DeepLTMLE as the base estimator (see Appendix C.6

for implementation details).

Figure 2 shows that both strategies improve over fully separate estimation, suggesting that sharing parameters across policies can reduce estimation variance. Notably, the multi-Q-head variant outperforms independent fine-tuning, indicating that jointly training within a unified model is more effective than adapting separate models after pretraining. Nevertheless, the proposed PEQ-Net achieves substantially lower RMSE than both alternatives. These results indicate that the improvement from PEQ-Net is not solely due to parameter-sharing, but arises from leveraging structured similarities between policies via the embedding.

**Joint training with dissimilar policies** PEQ-Net leverages structural similarity across policies to mitigate finite-sample variance inflation. To examine its behavior when this assumption is weakened, we additionally train PEQ-Net on policy sets containing extreme policies (thresholds 0 and 1), corresponding to "always treat" and "never treat", which are structurally dissimilar to intermediate threshold policies in the main results (Table 2). Experiment details are provided in Appendix C.2.

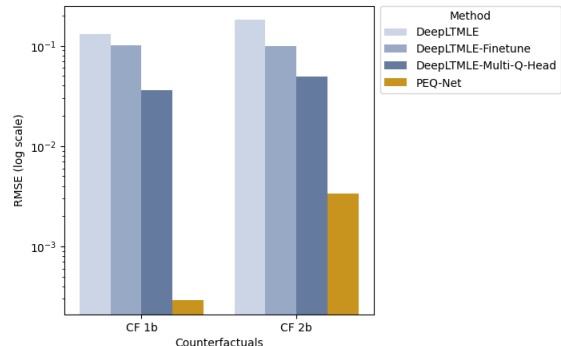

*Figure 2.* Ablation: PEQ-Net, Fine-tuning, Multi-Q-Head

*Table 3.* Performance under joint training with dissimilar policies. CF 1c, 2c, 3c, and 4c compare the baseline policy (threshold 0.5) against counterfactual policies with thresholds 0.4, 0.6, 0, and 1, respectively. PEQ-Net is trained on different policy sets to evaluate robustness to structurally dissimilar policies.

| Model | Bias | | RMSE | |
|---|---|---|---|---|
| | CF 1c | CF 2c | CF 1c | CF 2c |
| D.LTMLE | 0.30±0.22 | 0.43±0.25 | 0.37 | 0.50 |
| PEQ (joint: 0.4, 0.5, 0.6) | 0.11±0.09 | 0.25±0.10 | 0.14 | 0.27 |
| PEQ (joint: 0, 0.4, 0.5, 0.6, 1) | 0.14±0.11 | 0.11±0.09 | 0.18 | 0.14 |
| | CF 3c | CF 4c | CF 3c | CF 4c |
| D.LTMLE | 0.98±1.52 | 1.66±1.92 | 1.77 | 2.51 |
| PEQ (joint: 0, 0.5, 1) | 0.63±0.21 | 0.64±0.32 | 0.66 | 0.71 |
| PEQ (joint: 0, 0.4, 0.5, 0.6, 1) | 0.96±0.31 | 0.86±0.72 | 1.01 | 1.11 |

Table 3 shows that including both similar and dissimilar policies (i.e., $\{0, 0.4, 0.5, 0.6, 1\}$) does not consistently improve or degrade performance compared to smaller or more targeted configurations. Nevertheless, across all counterfactuals, PEQ-Net consistently outperforms DeepLTMLE under all training configurations. This suggests that the proposed policy embedding enables effective information sharing even when the policy set includes structurally dissimilar elements, providing robust gains over baselines without requiring careful selection of similar policies.

Overall, PEQ-Net is not limited to highly similar policies, but its advantage is the *strongest* when policies share sufficient structure. This aligns with our target application in retrospective policy evaluation and clinical hypothesis generation, where candidate policies are typically conservative and closely related. Grouping policies by similarity or adaptively controlling information sharing is an interesting direction for future work.

**Scalability**   The policy embedding in PEQ-Net relies on pairwise MMD computations, which introduce a quadratic dependence on both the number of policies $K$ and the sample size $N$. Let $d$ denote the feature dimension per time step and $\tau$ the horizon. The overall cost consists of: (i) an outer loop over policy pairs and time steps with complexity $O(K^2\tau)$, and (ii) an inner kernel-based MMD computation at each time step with cost $O(N^2 dt)$. In the worst case, this results in a total complexity of $O(K^2 N^2 \tau^2 d)$ for constructing the policy distance matrix.

Empirically, we observe a clear quadratic scaling with respect to the number of policies $K$ (Table C.5), consistent with the theoretical analysis. In practice, however, $K$ is typically moderate in clinical OPE settings ($K \ll N$), where policies correspond to a small number of expert-defined, interpretable treatment strategies. Moreover, the outer loop over policy pairs and time steps is also easily parallelizable.

The dependence on $N$ is the primary computational bottleneck, a limitation shared by kernel-based methods more broadly. A wide range of standard approximation techniques

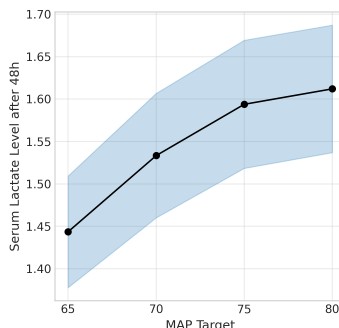

*Figure 3.* Higher MAP target associated with higher lactate level

Williams & Seeger (2000); Rahimi & Recht (2007); Rudi et al. (2017) can reduce the $O(N^2)$ complexity to near-linear or sub-quadratic complexity and can be incorporated into our framework.

### 5.4. Real-world Case Study

We applied PEQ-Net to a real-world cohort of sepsis patients with hypotension from the MIMIC-IV database to estimate the CATE of alternative vasopressor weaning strategies. The cohort includes 999 adult ICU patients who met Sepsis-3 criteria, developed hypotension (MAP $\leq$ 65 mmHg), and initiated vasopressor therapy within the first 24 hours of admission. The outcome of interest is serum lactate measured at 72 hours, a clinically relevant marker of tissue hypoperfusion in sepsis (Nguyen et al., 2010).

We considered four counterfactual policies defined by discontinuing vasopressors once MAP was maintained at or above 65, 70, 75, or 80 mmHg for 12 consecutive hours. Figure 3 reports CAPO estimates with 95% confidence intervals based on the estimated influence function. The results exhibit a consistent trend: higher MAP targets (more aggressive vasopressor use) are associated with higher lactate levels at 72 hours, although the estimated effects are modest. This pattern is consistent with a randomized control trial that reports no survival benefit from higher MAP targets (Asfar et al., 2014) and aligns directionally with current guideline recommendations to avoid unnecessary vasopressor exposure (Evans et al., 2021).

## 6. Conclusion

We identify a structural limitation of the prevailing separate estimation paradigm for longitudinal CATE estimation, in which independent nuisance fitting across counterfactual policies leads to finite-sample variance inflation. Leveraging the structure of ICE G-computation, we propose a policy-aware outcome reparameterization that enables principled model sharing across policies while preserving identification. Our approach yields a substantial reduction in RMSE.

## Impact Statement

This paper presents work whose goal is to advance the field of Machine Learning. There are many potential societal consequences of our work, none of which we feel must be specifically highlighted here.

## Acknowledgment

We thank Xintong Li for her contributions to MIMIC-III data processing and the reproduction of the MIMIC-Extract data extraction pipeline, and thank Zhenxing Xu for his contribution to MIMIC-IV data processing. We also thank Diyang Li for assistance with the theoretical derivations. This work utilized computing resources provided by AWS. We would also like to acknowledge the support from NSF 2212175, NIH RF1AG072449, RF1AG084178, R01AG080991, R01AG080624, R01AG076448, R01AG076234, and R01NS140142.

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

# A. Algorithms

## A.1. Inference

---
**Algorithm 3** PEQ-Net Inference
---

1: **Input:** Evaluation Data $\{O_i\}_{i=1}^m$; Prespecified counterfactual sequences $\{\boldsymbol{\pi}^{(k)}\}_{k=1}^K$; Learned model blocks $q, g, \phi, \rho$, with parameters $\theta$.
2: **Output:** $\hat{\psi}^{(k)}$ for $k = 1, \dots, K$
3: **for** $k = 1, \dots, K$ **do**
4:     **for** $t = 1$ to $\tau$ **do**
5:         $Q_t^{(k)}(A_t, H_t) = q(\phi(H_t, A_t); \tilde{\rho}(\rho_{t+1:\tau}^{(k)}))$
6:         $G_t = g(\phi(H_t))$
7:     **end for**
8:     $\hat{\psi}^{(k)} = LTMLE(\{O_i\}_{i=1}^m; \{Q_t^{(k)}\}_{t=1}^\tau; \{G_t\}_{t=1}^\tau)$
9: **end for**

---

## A.2. LTMLE

---
**Algorithm 4** LTMLE
---

1: **Input**: Dataset $\{O_i\}_{i=1}^n$; Initial estimates of $Q_t$ and $\hat{\pi}_t$ for $t = 1, \dots, \tau$; L1 regularization coefficient: $\lambda$.
2: **Output**: CAPO estimate $\hat{\psi}_n = \mathbb{P}_n Q_{1,\epsilon_1}(\dot{a}_1, H_1)$
3: Initialize $Q_{\tau+1} = Y, \epsilon_{\tau+1} = 0$
4: Initialize submodel:

$$\text{logit}Q_{t,\epsilon_t} = \text{logit}Q_t + \epsilon_t$$

and loss function

$$L_t(\epsilon_t) = \left( \prod_{s=1}^t \frac{\mathbb{1}(A_s = \dot{a}_s)}{\dot{a}_s \cdot \hat{\pi}_s(H_s) + (1 - \dot{a}_s)(1 - \hat{\pi}_s(H_s))} \right) \mathcal{L}_{bce}(Q_{t,\epsilon_t}(A_t, H_t), Q_{t+1,\epsilon_{t+1}}(\dot{a}_{t+1}, H_{t+1}))$$

5:
6: **for** $t = \tau$ to $1$ **do**
7:     $\epsilon_t \leftarrow \underset{\epsilon_t}{\arg \min} \mathbb{P}_n L_t(\epsilon_t) + \lambda|\epsilon_t|$
8: **end for**

---

# B. Theory and Proof

## B.1. Proof of Lemma 3.1

**Lemma 3.1.** *Consider the separate estimation paradigm where nuisance models are subject to independent random initialization. Even when two policies behave exactly the same at all steps, the second-order remainder in CATE generally does not cancel, i.e.,* $\text{Rem}^{(i),(j)} \neq 0$ *almost surely.*

*Proof.* **Notation.** For time $s$, let $Q_s^{(i),*}(A_s, H_s)$ and $\hat{Q}_s^{(i)}(A_s, H_s)$ denote the true and estimated outcome model under policy $\boldsymbol{\pi}^{(i)}$, and let $\pi_s^*(H_s)$ and $\hat{\pi}_s(H_s)$ denote the true and estimated propensity scores. Define the corresponding behavioral densities

$$\eta_s^*(A_s, H_s) = A_s \pi_s^*(H_s) + (1 - A_s)(1 - \pi_s^*(H_s)), \qquad \hat{\eta}_s(A_s, H_s) = A_s \hat{\pi}_s(H_s) + (1 - A_s)(1 - \hat{\pi}_s(H_s)).$$

The true behavioral density $\eta_s^*$ is invariant to the choice of counterfactual policy. However, under the separate estimation paradigm, nuisance models are refit independently for each policy. We therefore denote by $\hat{\eta}_s^{[i]}$ the behavioral density estimate obtained from the nuisance estimation run associated with policy $\boldsymbol{\pi}^{(i)}$.

For a counterfactual policy $\pi_s^{(i)}$, define the induced counterfactual density: $\eta_s^{(i)}(A_s, H_s) = \mathbb{1}\left(A_s = \pi_s^{(i)}(H_s)\right).$

To avoid ambiguity, note that $\hat{\eta}_s^{[i]}$ denotes an estimate of the behavioral density, whereas $\eta_s^{(i)}$ is fully determined by the counterfactual policy.

Define the following error and weight terms:

- Propensity error: $\xi_{G,s}^{[i]} = \hat{\eta}_s^{[i]} - \eta_s^*$,

- Outcome error: $\xi_{Q,s}^{(i)} = \hat{Q}_s^{(i)} - Q_s^{(i),*}$,

- Cumulative weight: $C_s^{(i)} = \prod_{r=1}^{s-1} \frac{\eta_r^{(i)}}{\hat{\eta}_r^{[i]}}$,

- Importance weight: $W_s^{(i)} = \frac{C_s^{(i)} \eta_s^{(i)}}{\eta_s^* \hat{\eta}_s^{[i]}} = \frac{1}{\eta_s^*} \prod_{r=1}^{s} \frac{\eta_r^{(i)}}{\hat{\eta}_r^{[i]}}.$

Analogous definitions apply for policy $\boldsymbol{\pi}^{(j)}$.

**Remainder representation.** By Lemma 1 of Díaz et al. (2023), the second-order remainder of the CAPO estimator under policy $\boldsymbol{\pi}^{(k)}$ admits the representation

$$\text{Rem}^{(k)} = \sum_{s=1}^{\tau-1} \mathbb{E}\left[R_s^{(k)}(A_s, H_s)\right],$$

where

$$R_s^{(k)} = C_s^{(k)} \left[\frac{\eta_s^{(k)}}{\hat{\eta}_s^{[k]}} - \frac{\eta_s^{(k)}}{\eta_s^*}\right] \left[\hat{Q}_s^{(k)} - Q_s^{(k),*}\right].$$

Since the CATE is defined as the difference between two CAPO estimators, its second-order remainder is

$$\text{Rem}^{(i),(j)} = \sum_{s=1}^{\tau-1} \mathbb{E}\left[R_s^{(i)}(A_s, H_s) - R_s^{(j)}(A_s, H_s)\right].$$

**Decomposition.** For a fixed time step $s$, algebraic manipulation yields

$$
\begin{aligned}
R_s^{(i)} - R_s^{(j)} &= C_s^{(i)} \cdot \left[\frac{\eta_s^{(i)}}{\hat{\eta}_s^{[i]}} - \frac{\eta_s^{(i)}}{\eta_s^*}\right] \cdot \left[\hat{Q}_s^{(i)} - Q_s^{(i),*}\right] - C_s^{(j)} \cdot \left[\frac{\eta_s^{(j)}}{\hat{\eta}_s^{[j]}} - \frac{\eta_s^{(j)}}{\eta_s^*}\right] \cdot \left[\hat{Q}_s^{(j)} - Q_s^{(j),*}\right] \\
&= \frac{C_s^{(i)} \eta_s^{(i)}}{\eta_s^* \hat{\eta}_s^{[i]}} \left[\eta_s^* - \hat{\eta}_s^{[i]}\right] \left[\hat{Q}_s^{(i)} - Q_s^{(i),*}\right] - \frac{C_s^{(j)} \eta_s^{(j)}}{\eta_s^* \hat{\eta}_s^{[j]}} \left[\eta_s^* - \hat{\eta}_s^{[j]}\right] \left[\hat{Q}_s^{(j)} - Q_s^{(j),*}\right] \\
&= W_s^{(i)} \xi_{G,s}^{[i]} \xi_{Q,s}^{(i)} - W_s^{(j)} \xi_{G,s}^{[j]} \xi_{Q,s}^{(j)} \\
&= W_s^{(i)} \xi_{G,s}^{[i]} \xi_{Q,s}^{(i)} - W_s^{(j)} \xi_{G,s}^{[i]} \xi_{Q,s}^{(i)} + W_s^{(j)} \xi_{G,s}^{[i]} \xi_{Q,s}^{(i)} - W_s^{(j)} \xi_{G,s}^{[j]} \xi_{Q,s}^{(j)} \\
&= (W_s^{(i)} - W_s^{(j)}) \xi_{G,s}^{[i]} \xi_{Q,s}^{(i)} + W_s^{(j)} (\xi_{G,s}^{[i]} \xi_{Q,s}^{(i)} - \xi_{G,s}^{[j]} \xi_{Q,s}^{(j)}) \\
&= (W_s^{(i)} - W_s^{(j)}) \xi_{G,s}^{[i]} \xi_{Q,s}^{(i)} + W_s^{(j)} (\xi_{G,s}^{[i]} \xi_{Q,s}^{(i)} - \xi_{G,s}^{[j]} \xi_{Q,s}^{(i)} + \xi_{G,s}^{[j]} \xi_{Q,s}^{(i)} - \xi_{G,s}^{[j]} \xi_{Q,s}^{(j)}) \\
&= \underbrace{(W_s^{(i)} - W_s^{(j)}) \xi_{G,s}^{[i]} \xi_{Q,s}^{(i)}}_{\text{Term I}} + \underbrace{W_s^{(j)} \xi_{Q,s}^{(i)} (\xi_{G,s}^{[i]} - \xi_{G,s}^{[j]})}_{\text{Term II}} + \underbrace{W_s^{(j)} \xi_{G,s}^{[j]} (\xi_{Q,s}^{(i)} - \xi_{Q,s}^{(j)})}_{\text{Term III}} \quad (7)
\end{aligned}
$$

**Non-vanishing under identical policies.** Assume $\boldsymbol{\pi}^{(i)} = \boldsymbol{\pi}^{(j)}$, so that $\eta_s^{(i)} = \eta_s^{(j)} = \eta_s$ for all $s$. Under the separate estimation paradigm, nuisance models are initialized and optimized independently, implying that

$$\hat{\eta}_s^{[i]} \neq \hat{\eta}_s^{[j]} \quad \text{and} \quad \hat{Q}_s^{(i)} \neq \hat{Q}_s^{(j)}$$

almost surely for at least one $s$.

We now show that each term above is non-zero almost surely.

- Term I: Since $\hat{\eta}_s^{[i]} \neq \hat{\eta}_s^{[j]}$, $W_s^{(i)} - W_s^{(j)} = \frac{\prod_{r=1}^s \eta_r}{\eta_s^*} \left( \prod_{r=1}^s \frac{1}{\hat{\eta}_r^{[i]}} - \prod_{r=1}^s \frac{1}{\hat{\eta}_r^{[j]}} \right) \neq 0$.

- Term II: $\xi_{G,s}^{[i]} - \xi_{G,s}^{[j]} = \hat{\eta}_s^{[i]} - \hat{\eta}_s^{[j]} \neq 0$.

- Term III: $\xi_{Q,s}^{(i)} - \xi_{Q,s}^{(j)} = \hat{Q}_s^{(i)} - \hat{Q}_s^{(j)} \neq 0$.

**Conclusion.** The second-order remainder of the CATE estimator is the sum of three non-vanishing terms. Although these terms depend on common nuisance estimates, they are functionally distinct. There is therefore no structural mechanism that enforces exact cancellation among them. Consequently,

$$\text{Rem}^{(i),(j)} \neq 0 \quad \text{almost surely,}$$

which completes the proof.

$\square$

### B.2. Proof of Lemma 4.1

**Lemma 4.1** (Identification of reparameterized $Q$-functions). *For any policy $\boldsymbol{\pi}^{(k)}$, the reparameterized outcome regressions $\{Q_t(\cdot, \cdot, \pi_{t+1:\tau}^{(k)})\}_{t=1}^{\tau}$ defined in Eq. (5) identify the same CAPO as standard ICE G-computation. In particular,*

$$\mathbb{E}\left[Q_1^{\text{ICE}}(\pi_1^{(k)}(H_1), H_1)\right] = \mathbb{E}\left[Q_1(\pi_1^{(k)}(H_1), H_1, \pi_{2:\tau}^{(k)})\right].$$

*Proof.* We show that the reparameterized $Q$-functions coincide pointwise with the standard ICE $Q$-functions, and therefore identify the same target parameter.

**Terminal step ($t = \tau$).** At the final time step, there is no future policy tail. By definition,

$$Q_\tau(A_\tau, H_\tau, \emptyset) = \mathbb{E}[Y \mid A_\tau, H_\tau] = Q_\tau^{\text{ICE}}(A_\tau, H_\tau), \tag{8}$$

so the two definitions agree.

**Inductive argument.** Fix a deterministic counterfactual policy $\boldsymbol{\pi}^{(k)}$. We proceed by backward induction on $t \in \{\tau - 1, \ldots, 1\}$.

*Base case ($t = \tau - 1$).* By the reparameterized definition,

$$\begin{aligned}
Q_{\tau-1}(A_{\tau-1}, H_{\tau-1}, \pi_\tau^{(k)}) &= \mathbb{E}\left[Q_\tau(\pi_\tau^{(k)}(H_\tau), H_\tau, \emptyset) \mid A_{\tau-1}, H_{\tau-1}, \pi_\tau^{(k)}\right] \\
&= \mathbb{E}\left[Q_\tau(\pi_\tau^{(k)}(H_\tau), H_\tau) \mid A_{\tau-1}, H_{\tau-1}\right] \\
&= \mathbb{E}\left[Q_\tau^{\text{ICE}}(\pi_\tau^{(k)}(H_\tau), H_\tau) \mid A_{\tau-1}, H_{\tau-1}\right] \\
&= Q_{\tau-1}^{\text{ICE}}(A_{\tau-1}, H_{\tau-1}).
\end{aligned}$$

The second equality follows because $\pi_\tau^{(k)}$ is deterministic and $Q_\tau(\cdot, \cdot)$ is the true outcome regression; conditioning on the fixed policy does not alter the conditional expectation given $(A_{\tau-1}, H_{\tau-1})$. The third equality uses Eq.(8).

*Inductive step.* Assume that for some $t + 1 \leq \tau - 1$,

$$Q_{t+1}(A_{t+1}, H_{t+1}, \pi_{t+2:\tau}^{(k)}) = Q_{t+1}^{\text{ICE}}(A_{t+1}, H_{t+1})$$

Then

$$\begin{aligned}
Q_t(A_t, H_t, \pi_{t+1:\tau}^{(k)}) &= \mathbb{E}\left[Q_{t+1}(\pi_{t+1}^{(k)}(H_{t+1}), H_{t+1}, \pi_{t+2:\tau}^{(k)}) \mid A_t, H_t, \pi_{t+1:\tau}^{(k)}\right] \\
&= \mathbb{E}\left[Q_{t+1}^{\text{ICE}}(\pi_{t+1}^{(k)}(H_{t+1}), H_{t+1}) \mid A_t, H_t, \pi_{t+1:\tau}^{(k)}\right] \\
&= \mathbb{E}\left[Q_{t+1}^{\text{ICE}}(\pi_{t+1}^{(k)}(H_{t+1}), H_{t+1}) \mid A_t, H_t\right] \\
&= Q_t^{\text{ICE}}(A_t, H_t),
\end{aligned}$$

where the second equality uses the inductive hypothesis and the third equality uses the determinism of $\pi_{t+1}^{(k)}$.

By induction, the two constructions coincide for all $t = 1, \ldots, \tau$. Evaluating at $(\dot{a}_1^{(k)}, H_1)$ and taking expectations yields the claimed equality of target parameters. $\qquad\square$

## B.3. Definition of Trajectory-level MMD

For a policy sequence $\pi_{1:t}$, the induced policy trajectory distribution is defined on the product space $\mathcal{Z}_1 \times \cdots \times \mathcal{Z}_t$. We equip this space with the product kernel $\prod_{r=1}^{t} k(\cdot, z_r)$, which induces the product RKHS $\mathcal{F}_{1:t} = \bigotimes_{r=1}^{t} \mathcal{F}_r$. The corresponding mean embedding can be represented as:

$$\mu_{1:t} = \mathbb{E}_{z_1 \sim P_1, \ldots, z_t \sim P_t}\Big[\prod_{r=1}^{t} k(\cdot, z_r)\Big] \in \mathcal{F}_{1:t}.$$

Distances between policy sequences can then be measured by the MMD between their trajectory-level mean embeddings, $\|\mu_{1:t}^{(i)} - \mu_{1:t}^{(j)}\|_{\mathcal{F}_{1:t}}$, computed analogously to the single-step case.

## B.4. MMD analysis of policy sequences

**Lemma B.1.** *Let $k$ be a normalized Gaussian kernel, i.e., $0 < k(\cdot, \cdot) \leq 1$ and $k(z, z) = 1$. For any subsequence interval $1 \leq m \leq n \leq \tau$, there exists a constant $C > 0$ such that*

$$\big\|\mu_{m:n}^{(i)} - \mu_{m:n}^{(j)}\big\|_{\mathcal{F}_{m:n}} \leq C \cdot \big\|\mu_{1:\tau}^{(i)} - \mu_{1:\tau}^{(j)}\big\|_{\mathcal{F}_{1:\tau}}.$$

*Proof.* Fix $1 \leq m \leq n \leq \tau$ and two policies $\boldsymbol{\pi}^{(i)}, \boldsymbol{\pi}^{(j)}$. Write $P_{1:\tau}^{(i)}$ and $P_{1:\tau}^{(j)}$ for the trajectory distributions of $Z_{1:\tau} \in \mathcal{Z}_1 \times \cdots \times \mathcal{Z}_\tau$ induced by $\boldsymbol{\pi}^{(i)}$ and $\boldsymbol{\pi}^{(j)}$, respectively, and let $P_{m:n}^{(i)}, P_{m:n}^{(j)}$ denote the corresponding marginals on $\mathcal{Z}_m \times \cdots \times \mathcal{Z}_n$. Let $\mu_{1:\tau}^{(i)}, \mu_{1:\tau}^{(j)} \in \mathcal{F}_{1:\tau}$ and $\mu_{m:n}^{(i)}, \mu_{m:n}^{(j)} \in \mathcal{F}_{m:n}$ be the associated mean embeddings under the product kernels.

We first note a uniform boundedness property implied by normalization. For any distribution $P$ on a space $\mathcal{Z}$ with RKHS $\mathcal{F}$ induced by $k$, the mean embedding $\mu_P = \mathbb{E}_{Z \sim P}[k(\cdot, Z)]$ satisfies

$$\|\mu_P\|_{\mathcal{F}}^2 = \big\langle \mathbb{E}_Z[k(\cdot, Z)], \mathbb{E}_{Z'}[k(\cdot, Z')]\big\rangle_{\mathcal{F}} = \mathbb{E}_{Z, Z' \sim P}\big[k(Z, Z')\big] \leq \mathbb{E}_{Z \sim P}\big[k(Z, Z)\big] = 1,$$

where the second equality follows from the reproducing property of the RKHS, and the inequality uses $k(Z, Z') \leq 1$. Thus $\|\mu_P\|_{\mathcal{F}} \leq 1$, and by the triangle inequality,

$$\|\mu_P - \mu_Q\|_{\mathcal{F}} \leq \|\mu_P\|_{\mathcal{F}} + \|\mu_Q\|_{\mathcal{F}} \leq 2 \qquad \text{for any } P, Q. \tag{9}$$

The same argument applies to the product RKHS $\mathcal{F}_{m:n}$ induced by the product kernel $k_{m:n}(z_{m:n}, z'_{m:n}) = \prod_{t=m}^{n} k_{\mathcal{Z}_t}(z_t, z'_t)$, since $0 < k_{m:n} \leq 1$ and $k_{m:n}(z, z) = 1$.

We now consider two cases.

**Case 1:** $\|\mu_{1:\tau}^{(i)} - \mu_{1:\tau}^{(j)}\|_{\mathcal{F}_{1:\tau}} = 0$. Because Gaussian kernels are characteristic, the product Gaussian kernel on $\mathcal{Z}_1 \times \cdots \times \mathcal{Z}_\tau$ is also characteristic, hence

$$\|\mu_{1:\tau}^{(i)} - \mu_{1:\tau}^{(j)}\|_{\mathcal{F}_{1:\tau}} = 0 \implies P_{1:\tau}^{(i)} = P_{1:\tau}^{(j)}.$$

Equality of the joint distributions implies equality of all marginals, so $P_{m:n}^{(i)} = P_{m:n}^{(j)}$. Applying characteristicness again on $\mathcal{Z}_m \times \cdots \times \mathcal{Z}_n$ yields

$$P_{m:n}^{(i)} = P_{m:n}^{(j)} \implies \|\mu_{m:n}^{(i)} - \mu_{m:n}^{(j)}\|_{\mathcal{F}_{m:n}} = 0.$$

Therefore the desired inequality holds for any $C > 0$ (e.g., take $C = 1$).

**Case 2:** $\|\mu_{1:\tau}^{(i)} - \mu_{1:\tau}^{(j)}\|_{\mathcal{F}_{1:\tau}} > 0$. Define

$$C := \frac{\|\mu_{m:n}^{(i)} - \mu_{m:n}^{(j)}\|_{\mathcal{F}_{m:n}}}{\|\mu_{1:\tau}^{(i)} - \mu_{1:\tau}^{(j)}\|_{\mathcal{F}_{1:\tau}}}.$$

Then by construction,

$$\|\mu_{m:n}^{(i)} - \mu_{m:n}^{(j)}\|_{\mathcal{F}_{m:n}} = C \cdot \|\mu_{1:\tau}^{(i)} - \mu_{1:\tau}^{(j)}\|_{\mathcal{F}_{1:\tau}}.$$

Moreover, $C$ is finite because the numerator is bounded by 2 via Eq. (9), while the denominator is strictly positive in this case. Hence, there exists a finite constant $C > 0$ satisfying the claimed inequality.

Combining the two cases completes the proof. $\qquad\square$

### B.5. Assumption

**Assumption B.2** (Regularity). We assume the true outcome model $Q_t^*$ and the counterfactual density product $\prod_{r=1}^t \eta_r$ (definition in Appendix B.1) are Lipschitz continuous with respect to the MMD between policy-induced distributions of their relevant sub-trajectory. That is, for any two policies $\boldsymbol{\pi}^{(i)}, \boldsymbol{\pi}^{(j)}$ and any time $t$, there exist constants $L_Q, L_\eta < \infty$ such that:

$$\|Q_t^{(i),*} - Q_t^{(j),*}\|_2 \leq L_Q \|\mu_{t+1:\tau}^{(i)} - \mu_{t+1:\tau}^{(j)}\|_{\mathcal{F}_{t+1:\tau}}, \tag{10}$$

$$\left\|\prod_{r=1}^t \eta_r^{(i)} - \prod_{r=1}^t \eta_r^{(j)}\right\|_2 \leq L_\eta \|\mu_{1:t}^{(i)} - \mu_{1:t}^{(j)}\|_{\mathcal{F}_{1:t}}. \tag{11}$$

### B.6. Proof of Theorem 4.2

**Theorem 4.2** (Lipschitz control of the CATE second-order remainder). *Suppose the parameterized outcome model and the policy encoder are Lipschitz in their respective inputs, and that the true outcome model and the policy-induced density product are Lipschitz continuous with respect to policy-induced trajectory distributions (detailed in Assumption B.2). Then there exists a finite constant $L_{\mathrm{R}} < \infty$ such that the magnitude of the second-order remainder in the CATE policy contrast is controlled by the trajectory-level policy distance:* $\left|\mathrm{Rem}^{(i),(j)}\right| \leq L_{\mathrm{R}} \|\mu_{1:\tau}^{(i)} - \mu_{1:\tau}^{(j)}\|_{\mathcal{F}_{1:\tau}}.$

*Proof.* We bound each component of $\mathrm{Rem}^{(i),(j)}$ in the decomposition(Eq. (7)).

- **Term I**: By Cauchy–Schwarz, $\left|(W_s^{(i)} - W_s^{(j)}) \xi_{G,s}^{[i]} \xi_{Q,s}^{(i)}\right| \leq \|W_s^{(i)} - W_s^{(j)}\|_2 \|\xi_{G,s}^{[i]} \xi_{Q,s}^{(i)}\|_2.$

  Using the shared estimate $\hat{\eta}_r$ and factoring common terms,

$$\|W_s^{(i)} - W_s^{(j)}\|_2 = \left\|\frac{1}{\eta_s^*}\prod_{r=1}^s \frac{\eta_r^{(i)}}{\hat{\eta}_r} - \frac{1}{\eta_s^*}\prod_{r=1}^s \frac{\eta_r^{(j)}}{\hat{\eta}_r}\right\|_2$$

$$= \left\|\frac{1}{\eta_s^*}\prod_{r=1}^s \frac{1}{\hat{\eta}_r}\left(\prod_{r=1}^s \eta_r^{(i)} - \prod_{r=1}^s \eta_r^{(j)}\right)\right\|_2$$

$$\leq \left\|\frac{1}{\eta_s^*}\prod_{r=1}^s \frac{1}{\hat{\eta}_r}\right\|_2 \left\|\prod_{r=1}^s \eta_r^{(i)} - \prod_{r=1}^s \eta_r^{(j)}\right\|_2.$$

By Assumption B.2 (Eq. (11)),

$$\left\|\prod_{r=1}^s \eta_r^{(i)} - \prod_{r=1}^s \eta_r^{(j)}\right\|_2 \leq L_\eta \|\mu_{1:s}^{(i)} - \mu_{1:s}^{(j)}\|_{\mathcal{F}_{1:s}}.$$

Applying Lemma B.1 yields $\|\mu_{1:s}^{(i)} - \mu_{1:s}^{(j)}\|_{\mathcal{F}_{1:s}} \leq C \|\mu_{1:\tau}^{(i)} - \mu_{1:\tau}^{(j)}\|_{\mathcal{F}_{1:\tau}}$. Therefore, there exists a finite constant $C_1$ such that

$$\left|(W_s^{(i)} - W_s^{(j)}) \xi_{G,s}^{[i]} \xi_{Q,s}^{(i)}\right| \leq C_1 \|\mu_{1:\tau}^{(i)} - \mu_{1:\tau}^{(j)}\|_{\mathcal{F}_{1:\tau}}.$$

- **Term II:** Under the shared modeling paradigm, the estimated treatment mechanism is identical across counterfactual policies, i.e., $\xi_{G,s}^{(i)} = \xi_{G,s}^{(j)}$. Hence this term vanishes:

$$W_s^{(j)} \xi_{Q,s}^{(i)} (\xi_{G,s}^{(i)} - \xi_{G,s}^{(j)}) = 0.$$

- **Term III:** The final LTMLE step updates the initial estimate $\hat{Q}_s^{(k)}$ to a targeted estimate $\hat{Q}_s^{(k),\epsilon}$ by solving the score equation. We note that under the strict positivity assumption, the fluctuation magnitude $\epsilon$ is bounded. Hence, this

bounded update preserves the local geometry of the initial estimate. Therefore, the continuity of the targeted estimate is controlled by the continuity of the initial neural network output:

$$\exists C_\epsilon < \infty, \|\hat{Q}_s^{(i),\epsilon_i} - \hat{Q}_s^{(j),\epsilon_j}\|_2 \leq C_\epsilon \|\hat{Q}_s^{(i)} - \hat{Q}_s^{(j)}\|_2. \tag{12}$$

This allows us to rely on the Lipschitz architecture of the initial $\hat{Q}$ to control Term III. Then, by Cauchy–Schwarz and the triangle inequality,

$$\left| W_s^{(j)} \xi_{G,s}^{(j)} (\xi_{Q,s}^{(i)} - \xi_{Q,s}^{(j)}) \right| \leq \|W_s^{(j)} \xi_{G,s}^{(j)}\|_2 \Big( \|\hat{Q}_s^{(i),\epsilon_i} - \hat{Q}_s^{(j),\epsilon_j}\|_2 + \|Q_s^{(i),*} - Q_s^{(j),*}\|_2 \Big)$$

$$\leq \|W_s^{(j)} \xi_{G,s}^{(j)}\|_2 \Big( C_\epsilon \|\hat{Q}_s^{(i)} - \hat{Q}_s^{(j)}\|_2 + \|Q_s^{(i),*} - Q_s^{(j),*}\|_2 \Big).$$

We next bound $\|\hat{Q}_s^{(i)} - \hat{Q}_s^{(j)}\|_2$ in terms of the trajectory-level policy distance. By assumption, the parameterized outcome model is Lipschitz in the policy embedding, $\tilde{\rho}(\rho_{s+1:\tau}^{(k)})$, we have

$$\|\hat{Q}_s^{(i)} - \hat{Q}_s^{(j)}\|_2 \leq L_{\hat{Q}} \|\tilde{\rho}(\rho_{s+1:\tau}^{(i)}) - \tilde{\rho}(\rho_{s+1:\tau}^{(j)})\|_2. \tag{13}$$

Moreover, the policy encoder $\tilde{\rho}$ is Lipschitz in its sequential inputs, so

$$\|\tilde{\rho}(\rho_{s+1:\tau}^{(i)}) - \tilde{\rho}(\rho_{s+1:\tau}^{(j)})\|_2 \leq L_{\tilde{\rho}} \sum_{r=s+1}^{\tau} \|\rho_r^{(i)} - \rho_r^{(j)}\|_2. \tag{14}$$

By construction of the per-time-step policy embedding, the embedding distance recovers the single-step MMD distance, i.e.,

$$\|\rho_r^{(i)} - \rho_r^{(j)}\|_2 = \|\mu_r^{(i)} - \mu_r^{(j)}\|_{\mathcal{F}_r} \qquad \text{for each } r \in \{1, \ldots, \tau\}. \tag{15}$$

Combining (13)–(15) yields

$$\|\hat{Q}_s^{(i)} - \hat{Q}_s^{(j)}\|_2 \leq L_{\hat{Q}} L_{\tilde{\rho}} \sum_{r=s+1}^{\tau} \|\mu_r^{(i)} - \mu_r^{(j)}\|_{\mathcal{F}_r}.$$

Finally, we lift each single-step distance to the full trajectory distance using Lemma B.1 with the interval $m = n = r$, which gives a constant $C$ such that for all $r$,

$$\|\mu_r^{(i)} - \mu_r^{(j)}\|_{\mathcal{F}_r} \leq C \|\mu_{1:\tau}^{(i)} - \mu_{1:\tau}^{(j)}\|_{\mathcal{F}_{1:\tau}}.$$

Therefore,

$$\|\hat{Q}_s^{(i)} - \hat{Q}_s^{(j)}\|_2 \leq L_{\hat{Q}} L_{\tilde{\rho}} (\tau - s) C \|\mu_{1:\tau}^{(i)} - \mu_{1:\tau}^{(j)}\|_{\mathcal{F}_{1:\tau}}. \tag{16}$$

For the true outcome model difference $\|Q_s^{(i),*} - Q_s^{(j),*}\|_2$, Assumption B.5 (Eq. (10)) together with Lemma B.1 yields

$$\|Q_s^{(i),*} - Q_s^{(j),*}\|_2 \leq L_Q \|\mu_{s+1:\tau}^{(i)} - \mu_{s+1:\tau}^{(j)}\|_{\mathcal{F}_{s+1:\tau}} \leq L_Q C \|\mu_{1:\tau}^{(i)} - \mu_{1:\tau}^{(j)}\|_{\mathcal{F}_{1:\tau}}.$$

Substituting the above bounds into the display at the beginning of Term III shows that

$$\left| W_s^{(j)} \xi_{G,s}^{(j)} (\xi_{Q,s}^{(i)} - \xi_{Q,s}^{(j)}) \right| \leq C_3 \|\mu_{1:\tau}^{(i)} - \mu_{1:\tau}^{(j)}\|_{\mathcal{F}_{1:\tau}}$$

for some finite constant $C_3$.

**Conclusion.** Summing over $s$ in Eq. (7), Term II is identically zero and Terms I and III are each bounded by a finite constant multiple of $\|\mu_{1:\tau}^{(i)} - \mu_{1:\tau}^{(j)}\|_{\mathcal{F}_{1:\tau}}$. Therefore, there exists $L_{\mathrm{R}} < \infty$ such that

$$|\mathrm{Rem}^{(i),(j)}| \leq L_{\mathrm{R}} \|\mu_{1:\tau}^{(i)} - \mu_{1:\tau}^{(j)}\|_{\mathcal{F}_{1:\tau}},$$

which completes the proof.

$\square$

## C. Experiment Setting

### C.1. Semi-Synthetic Data DGP

**Limited Semi-synthetic DGP**  We adopt the semi-synthetic DGP from DeepACE. Using the preprocessing pipeline of Wang et al. (2020), we extract 10 time-varying covariates $X_t \in \mathbb{R}^{10}$ over $\tau = 15$ time steps. These covariates are treated as observed patient states and are directly taken from real-world measurements.

Given the observed history, binary treatments $A_t \in \{0, 1\}$ are simulated according to a stochastic, dynamic treatment policy. The treatment assignment probability at time $t$ depends on past covariates, past intermediate outcomes, and a treatment intensity variable $\ell_t = \ell_{t-1} + 2(A_t - 1)\bar{X}_t \tanh(Y_t)$, with initialization $\ell_0 = T/2 - 3$. Treatment is assigned as

$$\pi_t = \mathbb{1}\left\{\sigma\left(\sum_{i=1}^{h} \frac{(-1)^i}{1-i}\left(\bar{X}_{t-i} + \tanh(Y_{t-i}/2)\right) - \tanh\left(\ell_{t-1} - \frac{T}{2}\right) + \epsilon_t^A\right) > 0.5\right\},$$

and outcomes are generated according to

$$Y_{t+1} = 5\sum_{i=1}^{h} \frac{(-1)^i}{1-i}\tanh\left(\sin(\bar{X}_{1:5,t-i}A_{t-i}) + \cos(\bar{X}_{5:10,t-i}A_{t-i})\right) + \epsilon_t^Y,$$

where $\bar{X}_{1:5,t-i}$ represent the mean of the first five covariates at step $t - i$, similarly for $\bar{X}_{5:10,t-i}$. The time lag $h = 8$. Noise terms satisfy $\epsilon_t^A, \epsilon_t^Y \sim \mathcal{N}(0, 0.5^2)$.

Our target estimand is the terminal outcome $Y_{15}$. Intermediate outcomes $Y_t$ for $t < 15$ are absorbed into $L_{t+1}$ (Section 2), yielding $L_t = (X_t, Y_{t-1})^\top \in \mathbb{R}^{11}$.

**Expanded Semi-synthetic DGP**  In the simple DGP above, patient covariates are unaffected by treatment. To introduce a stronger treatment–confounder feedback loop, we further synthesize treatment-dependent time-varying covariates.

Specifically, we generate latent covariates $Z_t \in \mathbb{R}^5$ with initialization $Z_1 \sim \mathcal{N}(0, I)$ and dynamics

$$Z_{t+1} = \omega_1 \cdot Z_t + \omega_2 \cdot A_t \sigma(Z_t^2) + \omega_3 \cdot \bar{X}_t + \epsilon_t^Z,$$

where $\epsilon_t^Z \sim \mathcal{N}(0, 0.3^2)$ and the coefficients $\omega = (\omega_1, \omega_2, \omega_3)$ are randomly drawn and fixed to (0.37, 0.42, 0.29). These synthesized covariates both affect and are affected by treatment.

We concatenate $Z_t$ with the observed covariates to form the augmented state

$$X_t^\dagger = (X_t, Z_t)^\top \in \mathbb{R}^{15}.$$

Treatment assignment and outcome generation follow the same structural equations as in the Limited Semi-synthetic DGP, with $X_t$ replaced by $X_t^\dagger$. Noise distributions remain unchanged.

### C.2. Counterfactuals

We consider two types of counterfactual treatment regimes:

- (a) Deterministic treatment sequences. When counterfactuals correspond to fixed binary treatment values at each time step, we adapt the setup of DeepACE. The baseline policy is the *always-treat* policy, under which treatment is always administered (i.e., $A_t = 1$ for all $\tau = 15$ time steps). In addition to this baseline, we evaluate three deterministic counterfactual sequences: (3) treatment is never administered at any time step; (2) treatment is initiated at time step 5 and continued until the end; and (3) treatment is administered during the first 10 steps and then discontinued. In results, we use CF 1a, CF 2a, and CF 3a to denote the CATE between the always-treat policy and the three counterfactual policies, respectively.

- (b) Dynamic treatment policies. We consider deterministic, time-varying treatment policies in experiments. These policies are constructed by varying the threshold in the treatment assignment rule while keeping the functional form

fixed. Specifically, for the limited semi-synthetic DGP, we define a policy sequence $\boldsymbol{\pi} = (\pi_1, \ldots, \pi_\tau)$, where each decision rule $\pi_t$ is parameterized by a threshold $\gamma$:

$$\pi_t = \mathbb{1}\left\{\sigma\left(\sum_{i=1}^{h} \frac{(-1)^i}{1-i}\left(\bar{X}_{t-i} + \tanh(Y_{t-i}/2)\right) - \tanh\left(\ell_{t-1} - \frac{T}{2}\right)\right) > \gamma\right\}. \tag{17}$$

The policy in the expanded DGP also follows the same functional form, but the covariates $X_t$ are replaced by the augmented covariates $X_t^\dagger$.

In both the limited and expanded semi-synthetic DGPs, we use the policy $\gamma = 0.5$ across all time steps as the baseline policy. All reported results correspond to contrasts between each counterfactual policy and this baseline.

We consider two scenarios with different levels of policy similarity:

(1) *Partial deviation:* counterfactual policies differ from the baseline only in the first two time steps. Specifically, CF 1b corresponds to the contrast between the baseline policy and a policy with $\gamma = 0.4$ for the first two steps and $\gamma = 0.5$ thereafter, while CF 2b corresponds to $\gamma = 0.6$ for the first two steps and $\gamma = 0.5$ thereafter. In other words, these counterfactuals share identical actions with the baseline for the last 13 steps.

(2) *Full deviation:* counterfactual policies differ from the baseline at all time steps. CF 1c and CF 2c correspond to the contrasts with constant policies $\gamma = 0.4$ and $\gamma = 0.6$, respectively. In the ablation study, we further include two extreme policies, with CF 3c and CF 4c corresponding to $\gamma = 0$ ("always treat") and $\gamma = 1$ ("never treat") at all time steps.

### C.3. Model Implementation

We implement G-computation and LTMLE with Super Learner using the `ltmle` R package, setting `gcomp=TRUE` for the former. Super Learner constructs an optimally weighted ensemble of base learners via $k$-fold cross-validation; throughout, we set $k = 3$ and include a generalized linear model, random forest, XGBoost, and a generalized additive model (GAM) with regression splines as candidate learners.

DeepACE is implemented using the authors' original codebase (Frauen et al., 2023). For DeepLTMLE, we follow the original paper to derive our own implementation.

For PEQ-Net, we extend DeepLTMLE with our policy encoder module. To construct policy embedding, we compute the pairwise MMD distance using Gaussian kernel, and apply metric MDS from `scikit-learn` to obtain the per-time-step policy encoding. The kernel bandwidth is selected using a median heuristic: at each time step, we randomly sample 500 history–action pairs, compute all pairwise Euclidean distances, and take their median; the bandwidth is then set to $1/(2 * \text{median})$. The number of layers and hidden dimensions of the policy encoder (implemented as an RNN) are treated as hyperparameters; here, both the Transformer block and the policy encoder share the hyperparameter 'number of layers'.

All deep learning models, including DeepACE, DeepLTMLE, and our proposed method, are trained for 500 epochs.

For all baseline models, including G-computation and LTMLE with super learner, DeepACE, and DeepLTMLE, we follow their original implementation and fit separate models to estimate the CAPO for each counterfactual. For our model, we fit one model to estimate the CAPO for all counterfactuals. Finally, we take the difference between these CAPOs to derive CATE estimates.

### C.4. Hyperparameter Tuning

We use the random search strategy to tune the hyperparameters for all models. Since we only have factual data, we select hyperparameter by minimizing the factual loss: $L_{mse}(Q(A_\tau, H_\tau), Y) + \sum_{t=1}^{\tau} L_{bce}(G(H_t), A_t)$. The hyperparameter grid is shown in Table C.1.

### C.5. Ablation: ICE G-computation only

In this ablation study, we followed the exact same strategy in Section C.4 to train DeepLTMLE and PEQ-Net on the same datasets. After training, we don't conduct the LTMLE debiasing step but directly take the average of $Q_1(\pi(H_1), H_1)$ as the CAPO estimates, following the original ICE G-computation procedure. Results are shown in Table C.2 and C.3.

*Table C.1.* Hyperparameter tuning grid used for all datasets and counterfactual settings.

| Hyperparameter | DeepACE | DeepLTMLE | PEQ-Net |
|---|---|---|---|
| Batch size | {128, 256} | {128, 256} | {128, 256} |
| Learning rate | {5e-4, 1e-3, 5e-3} | {5e-4, 1e-3, 5e-3} | {5e-4, 1e-3, 5e-3} |
| Hidden size | {8, 16, 32} | {8, 16, 32} | {8, 16, 32} |
| Dropout rate | {0.0, 0.1} | {0.0, 0.1} | {0.0, 0.1} |
| Number of layers | — | {1, 2, 3} | {1, 2, 3} |
| Number of heads | — | {2, 4} | {2, 4} |
| $\alpha$ | — | {0.05, 0.1} | {0.05, 0.1} |
| Hidden size of policy encoder | — | — | {8,16} |

*Table C.2.* Ablation: Isolating Improvements in the Outcome Regression ($Q$) with counterfactuals of dynamic treatment policy.

| DGP | Model | Policy differ only in the first two steps | | | | Policy differ in all steps | | | |
|---|---|---|---|---|---|---|---|---|---|
| | | Bias | | RMSE | | Bias | | RMSE | |
| | | CF 1b | CF 2b | CF 1b | CF 2b | CF 1c | CF 2c | CF 1c | CF 2c |
| Limited | D.LTMLE | 0.0998±0.0780 | 0.1007±0.1070 | 0.1255 | 0.1450 | 0.2316 ± 0.1522 | 0.2335 ±0.1423 | 0.2750 | 0.2716 |
| | PEQ-Net | **0.0015±0.0021** | **0.0132±0.0070** | **0.0026** | **0.0148** | **0.0957 ±0.0770** | **0.1410 ± 0.0746** | **0.1216** | **0.1586** |
| Expanded | D.LTMLE | 0.0906±0.0887 | 0.1270±0.1449 | 0.1252 | 0.1899 | 0.3751 ± 0.2970 | 0.4883 ± 0.3064 | 0.4738 | 0.5724 |
| | PEQ-Net | **0.0003±0.0001** | **0.0030±0.0017** | **0.0003** | **0.0034** | **0.0916±0.0569** | **0.2726±0.0881** | **0.1071** | **0.2858** |

## C.6. Ablation: Comparison with finetuning and multi-Q-head strategy of DeepLTMLE

This ablation study is conducted on the counterfactuals with dynamic treatment policies using the dataset with expanded time-varying confounding.

For finetuning, we first train the DeepLTMLE model for the behavioral policy (threshold=0.5 for all $\tau = 15$ time steps) on the combined training set, yielding the CAPO for this sequence. Then we freeze the parameters in the transformer block and the $G$-head but leave the $Q$-head trainable, and train the model on the remaining counterfactual sequence one by one.

For the multi-Q-head architecture, we modify DeepLTMLE: the transformer block and the $G$-head are unchanged, and multiple $Q$-heads are initiated, one for each policy. The model is jointly trained on all policies.

Finally, we obtain CATE by differencing the CAPOs. These experiments were repeated on data generated with 20 different random seeds, and the mean absolute bias is reported.

## C.7. Additional analysis on policy embedding

**Visualization of policy embedding** To inspect whether our policy embedding procedure can reflect the similarity among policies, we embed five counterfactual policies with different thresholds $\gamma$ (see Eq (17) for their functional form) across 15 time steps. Figure C.1 shows a consistent geometric structure across time steps: policies with similar intervention patterns are mapped to nearby representations, while more dissimilar policies are clearly separated. This indicates that the embedding captures meaningful relationships between policies in a structured manner.

**Sensitivity to kernel bandwidth choice** In our policy embedding, the kernel bandwidth is selected using a median heuristic: at each time step, we randomly sample 500 history–action pairs, compute all pairwise Euclidean distances, and take their median; the bandwidth is then set to $1/(2 * \text{median})$. For sensitivity analysis, we vary it as $\gamma/(2 * \text{median})$ with $\gamma \in \{0.01, 0.1, 1, 10, 100\}$. Table C.4 shows that RMSE on the expanded DGP remains stable across settings, indicating PEQ-Net's performance is robust to the bandwidth choices.

**Scalability with respect to the number of policies** We run the policy embedding procedure with a varying number of policies $K$. Table C.5 confirms its quadratic computational complexity with respect to $K$.

*Table C.3.* Ablation: Isolating Improvements in the Outcome Regression ($Q$) with counterfactuals of deterministic sequences.

| Model | Limited Time-varying Confounding | | | | | | Expanded Time-varying Confounding | | | | | |
|---|---|---|---|---|---|---|---|---|---|---|---|---|
| | Bias | | | RMSE | | | Bias | | | RMSE | | |
| | CF 1a | CF 2a | CF 3a | CF 1a | CF 2a | CF 3a | CF 1a | CF 2a | CF 3a | CF 1a | CF 2a | CF 3a |
| D.LTMLE ($Q$ only) | 0.78±0.65 | 0.68±0.56 | 0.92±0.95 | 1.00 | 0.88 | 1.31 | 0.80±0.76 | 1.10±0.85 | 0.58±0.58 | 1.09 | 1.37 | 0.81 |
| PEQ-Net ($Q$ only) | **0.53±0.38** | **0.27±0.24** | **0.21±0.19** | **0.65** | **0.36** | **0.28** | **0.63±0.36** | **0.20±0.17** | **0.26±0.22** | **0.72** | **0.26** | **0.33** |

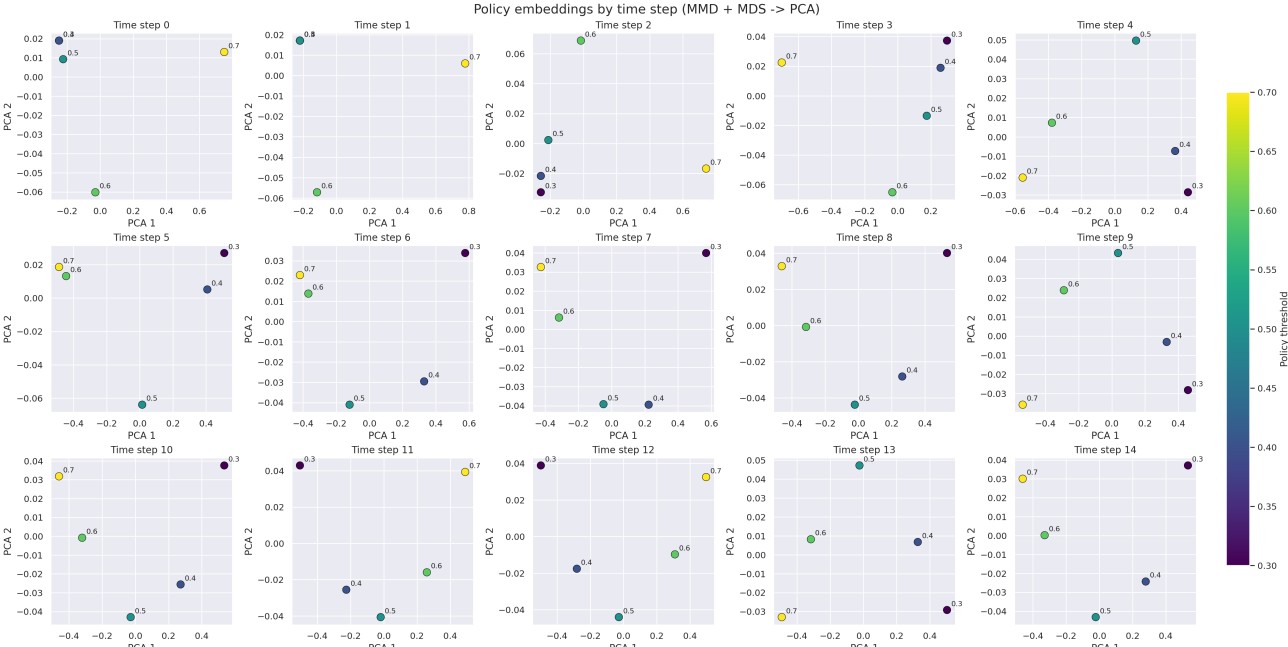

*Figure C.1.* Visualization of policy embedding across time steps via PCA

*Table C.4.* PEQ-Net with varying kernel bandwidth for policy embedding. Results are aggregated over 10 seeds.

| $\gamma$ | RMSE | |
|---|---|---|
| | CF 1c | CF 2c |
| 0.01 | 0.21 | 0.23 |
| 0.1 | 0.22 | 0.27 |
| 1 | 0.17 | 0.23 |
| 10 | 0.21 | 0.24 |
| 100 | 0.21 | 0.21 |

*Table C.5.* Runtime of policy embedding with respect to the number of policies $K$.

| $K$ | Runtime |
|---|---|
| 3 | 2.68 s |
| 5 | 4.36 s |
| 10 | 13.09 s |
| 20 | 53.18 s |

