# OpenReview forum: "Smooth Multi-Policy Causal Effect Estimation in Longitudinal Settings"
_ICML.cc/2026/Conference — ICML 2026 regular_

### Official Review · Reviewer_kNin · 2026-03-06

**Soundness:** 3
**Presentation:** 3
**Significance:** 3
**Originality:** 3
**Overall Recommendation:** 4
**Confidence:** 3

**Summary:**

This paper studies the estimation of potential outcomes in longitudinal settings, where a policy predicts actions at each time step based on the historical actions and covariates. The paper considers multiple counterfactual policies jointly, rather than a single counterfactual policy as in prior research. The main idea is to modify the previous method so that the Q-model is explicitly conditioned on the future policy tail and trained jointly across all counterfactual policies. The proposed PEQ-Net is then empirically evaluated on semi-synthetic MIMIC-III–derived datasets and a real-world MIMIC-IV vasopressor weaning case study.

**Compliance With Llm Reviewing Policy:**

Affirmed.

**Final Justification:**

I will maintain my score based on the rebuttal.

**Key Questions For Authors:**

In your view, how does the degree of policy dissimilarity affect these gains, and is there an expected "sweet spot" where policies are neither too similar nor too different?

**Limitations:**

Yes

**Strengths And Weaknesses:**

The paper identifies a clear gap and tackles a well-defined problem that is appropriately situated within the existing literature. Its use of RKHS mean embeddings and MMD to quantify population-level policy dissimilarity is a reasonable and viable design choice. The proposed reparameterization of the ICE Q-function, which makes the dependence on the future policy tail explicit, is logically coherent. Empirically, the results consistently demonstrate the proposed method’s advantages over the considered baselines.

However, I believe the paper misses an important and potentially insightful discussion. The empirical results suggest that estimating three counterfactual policies jointly yields higher accuracy than estimating each policy in isolation, which is reminiscent of the benefits observed in multi-task learning. At the same time, such gains should plausibly depend on how related the policies are: if the policies are too dissimilar, there may be little shared structure to exploit, whereas if the policies are identical, joint estimation should provide little to no additional benefit. This naturally raises the question of how policy dissimilarity affects performance. It would be valuable to include at least a small empirical study that varies the degree of dissimilarity among policies and examines its impact on estimation accuracy.

---

> ### Author Rebuttal · Authors · 2026-03-31
>
> We thank the reviewer for raising this important and insightful discussion. To study the behavior under dissimilar policies, we design additional experiments where the model is jointly trained on policy sets that include extreme policies (0 and 1 as thresholds), corresponding to “always treat” and “never treat,” which are structurally dissimilar to intermediate threshold policies.
>
> We consider three settings: joint training with thresholds (1) {0.4, 0.5, 0.6}, (2) {0, 0.5, 1}, and (3) {0, 0.4, 0.5, 0.6, 1}. We evaluate CATE relative to the behavioral policy (0.5), with results shown below, where CF 1d, 2d, 3d, 4d denote the other four policies (0, 0.3, 0.7, 1), respectively.
>
> | Method     | Bias(CF 2d) | Bias(CF 3d) | RMSE(CF 2d) | RMSE(CF 3d) |
> |------------| ------------|-------------|-------------|-------------|
> |D.LTMLE                           |0.26±0.19|0.32±0.27| 0.31 | 0.42 |
> |PEQ-Net(joint:0.4, 0.5, 0.6)      |0.13±0.09|0.28±0.07| 0.16 | 0.29 |
> |PEQ-Net(joint:0, 0.4, 0.5, 0.6, 1)|0.11±0.07|0.11±0.09| 0.12 | 0.14 |
>
> | Method     | Bias(CF 1d) | Bias(CF 4d) | RMSE(CF 1d) | RMSE(CF 4d) |
> |------------| ------------|-------------|-------------|-------------|
> |D.LTMLE|                           |0.39±0.39| 0.41±0.61| 0.55 | 0.72 |
> |PEQ-Net (joint:0, 0.5, 1)          |0.21±0.17| 0.37±0.24| 0.27 | 0.44 |
> |PEQ-Net (joint:0, 0.4, 0.5, 0.6, 1)|0.52±0.23| 0.42±0.35| 0.56 | 0.54 |
>
>
> First, for similar policies {0.4,0.5,0.6}, joint training substantially reduces bias and RMSE for intermediate counterfactuals (CF 2d, 3d) compared to D.LTMLE, showing clear gains when policies are close.
>
> Second, for highly dissimilar policies {0,0.5,1}, joint training still improves performance for extreme policies (CF 1d, 4d), indicating that PEQ-Net can exploit shared structure beyond very local neighborhoods.
>
> Third, when combining both similar and dissimilar policies {0,0.4,0.5,0.6,1}, performance further improves for intermediate counterfactuals, while remaining comparable for extreme ones (e.g., CF 1d), suggesting negative transfer.
>
> Overall, these results indicate that PEQ-Net is not restricted to only very similar policies, but its advantage is **strongest** when the policies share sufficient structure. This also aligns with our target application—retrospective policy evaluation and clinical hypothesis generation. In clinical settings, candidate policies are usually conservative and clinically plausible, so one typically compares a small set of related strategies rather than drastically different ones. Thus, while extreme dissimilarity is an important boundary case, the practically relevant regime is one where policies remain reasonably close and joint estimation is most useful. Grouping policies by similarity or adaptively controlling information sharing would be interesting directions for future work.

---

> > ### Author Rebuttal · Reviewer_kNin · 2026-04-03
> >
> > I appreciate the authors' responses. I will keep my positive score.

---

### Official Review · Reviewer_W6L6 · 2026-03-06

**Soundness:** 3
**Presentation:** 3
**Significance:** 3
**Originality:** 3
**Overall Recommendation:** 5
**Confidence:** 3

**Summary:**

This paper investigates the problem of variance inflation when evaluating multiple dynamic treatment policies in longitudinal causal inference, focusing on how the conventional separate estimation paradigm induces structurally uncontrolled second-order bias. The authors propose the Policy-Encoded Q Network (PEQ-Net), which addresses this issue by introducing a policy-aware reparameterization of Iterative Conditional Expectation (ICE) Q-functions and utilizing RKHS-based policy embeddings to enable joint estimation and shared representations across counterfactuals. They validate their method theoretically, by proving it imposes a structural constraint on the second-order remainder after LTMLE debiasing, and empirically on semi-synthetic MIMIC datasets and a real-world sepsis case study, demonstrating that PEQ-Net significantly reduces root-mean-square error compared to existing ICE-based baselines.

**Compliance With Llm Reviewing Policy:**

Affirmed.

**Final Justification:**

This paper investigates the important problem of variance inflation when evaluating multiple dynamic treatment policies and proposes PEQ-Net for joint estimation. My main concerns were related to the computational scalability of the MMD distance construction and how the method handles highly dissimilar policies. In their rebuttal, the authors provided a clear runtime analysis showing the computational cost is manageable in practice, and they added new ablation studies demonstrating the model's robustness even when extreme policies are included. These clarifications and additional experiments effectively addressed my concerns. Since I find the research problem highly meaningful and the proposed solution solid, I am willing to raise my score from 4 to 5.

**Key Questions For Authors:**

Please check the Weaknesses.

**Limitations:**

yes

**Strengths And Weaknesses:**

**Strengths:**

- The paper addresses a critical and practical bottleneck in longitudinal causal inference: the finite-sample variance inflation caused by the conventional separate estimation paradigm.

- The proposed approach is highly novel. It creatively integrates RKHS-based policy embeddings into the ICE G-computation pipeline, enabling a single network to share representations across multiple counterfactual policies.

- The theoretical foundation is solid, and empirical results on both semi-synthetic MIMIC data and a real-world sepsis case study clearly demonstrate the effectiveness of PEQ-Net.

**Weaknesses:**

- Computational Scalability: The method requires pre-computing distances (MMD and MDS) between all candidate policies . I am concerned about whether the computational time will increase significantly as the number of patient trajectories or the number of policies grows, which might limit its scalability to larger datasets.

- Applicability to Dissimilar Policies: While the joint representation is effective for information sharing, I am curious about its behavior when the policies being compared are highly dissimilar. It would be helpful to understand if there are potential trade-offs or boundary cases where the joint estimation might face challenges compared to the traditional separate estimation paradigm.

---

> ### Author Rebuttal · Authors · 2026-03-31
>
> **W1:** We thank the reviewer for raising the scalability concern regarding the MMD step. We agree that the pairwise computation of MMD introduces a quadratic dependence on the number of policies $K$ and the sample size $N$.
>
> Let $d$ denote the feature dimension per time step; our computation cost consists of (1) an outer loop over policy pairs and time steps, yielding $O(K^2\tau)$, and (2) an inner kernel-based MMD computation with cost $O(N^2dt)$ at time step $t$. Together, this results in a worst-case computational complexity of $O(K^2N^2\tau^2d)$ for constructing the distance matrix. We empirically validate the quadratic dependence on $K$:
> |$K$|Runtime(s)|
> |---|----------|
> | 3 | 2.68s    |
> | 5 | 4.36s    |
> |10 | 13.09s   |
> |20 | 53.18s   |
>
> *On the role of $K$:* For off policy evaluation in clinical care, $K$ is typically moderate and not the dominant factor ($K\ll N$). This is because policies are pre-specified by domain experts and correspond to a small set of interpretable and actionable strategies (e.g., a few MAP target thresholds), rather than a large or dense policy set. Additionally, the $O(K^2 \tau)$ outer loop can be parallelized across cores, which effectively mitigates the computational burden in practice.
>
> *On the role of $N$:* We agree that the quadratic complexity in $N$ is the primary computational bottleneck. This is a general challenge shared by kernel-based methods. A wide range of standard approximation techniques can be applied here to mitigate the cost, such as [1-3]. These approaches can reduce the $O(N^2)$ complexity to near-linear or sub-quadratic while preserving accuracy, and can be incorporated into our framework.
>
> [1] Williams, Christopher, and Matthias Seeger. "Using the Nyström method to speed up kernel machines." Advances in neural information processing systems 13 (2000).
> [2] Rahimi, Ali, and Benjamin Recht. "Random features for large-scale kernel machines." Advances in neural information processing systems 20 (2007).
> [3] Rudi, Alessandro, Luigi Carratino, and Lorenzo Rosasco. "Falkon: An optimal large scale kernel method." Advances in neural information processing systems 30 (2017).
>
>
> **W2** We thank the reviewer for this insightful suggestion. We refer the reviewer to our response to kNin for an ablation study and a detailed discussion on incorporating dissimilar policies.

---

> > ### Author Rebuttal · Reviewer_W6L6 · 2026-04-01
> >
> > Thank you for the detailed rebuttal and additional experiments. The clarifications effectively answered my questions, and I have raised my rating accordingly.

---

### Official Review · Reviewer_f8ah · 2026-03-07

**Soundness:** 2
**Presentation:** 2
**Significance:** 2
**Originality:** 3
**Overall Recommendation:** 3
**Confidence:** 4

**Summary:**

The paper introduces a framework for longitudinal causal effect estimation when evaluating multiple dynamic treatment policies. The authors point out a critical flaw that estimating Q-functions for different policies in isolation leads to uncontrolled second-order bias when computing CATE. Hence, to solve the issue, the authors propose the policy-encoded Q network. It involves mapping policy-induced actions into a continuous representation space . These policy embeddings are then fed into a shared Transformer-based Q network.

**Compliance With Llm Reviewing Policy:**

Affirmed.

**Final Justification:**

Please see the acknowledgement comment

**Key Questions For Authors:**

1. Could you provide a more convincing justification for the assumption that the true conditional outcome is Lipschitz continuous with respect to the MMD of policy distributions? Can the model effectively navigate the complexity of clinical care, particularly when minor adjustments in intervention timing lead to different patient outcomes?
2. What is the computational complexity of generating the per-time-step policy embedding? How does PEQ-Net scale when the number of counterfactual policies K is in the thousands or when policies are continuous rather than discrete?
3. Could you please explain how sensitive the final CATE estimation is to the choice of the characteristic kernel and its bandwidth during the RKHS embedding step? please provide a sensitivity analysis on this.
4. The fine-tuning ablation is appreciated. A standard approach in deep learning for multiple outputs is a multi-head architecture. I wonder how PEQ-Net compare empirically against the simpler form of structural sharing?

**Limitations:**

1. The authors do not adequately discuss the limitations of their work. The work relies on the assumption of sequential ignorability (no unmeasured confounding). This assumption is challenged by clinicians' reliance on other qualitative cues not captured in electronic health records. The authors should discuss how sensitive PEQ-Net is to unmeasured confounders.
2. The model uses a pairwise MMD distance matrix, its computational complexity could scales quadratically relative to the number of policies.
3. The method relies on positivity assumption heavily. But in reality, if we force an unusually high MAP target, there simply won't be enough patients in the observational data who actually received that exact treatment. The model might end up guessing in the unsupported areas all while presenting its estimates with high confidence.
4. Putting off-policy evaluation models into the environment like ICUs could be risky. There might be some real issues with automation bias where busy doctors might just blindly trust the AI's causal estimates without questioning them.

**Strengths And Weaknesses:**

1. Soundness:
(1) Identifying the non-canceling second-order remainder in the separate estimation paradigm is mathematically sound and shows a structural inefficiency in current methods.
(2) Assumption B2 assumes that the true outcome model $Q^*_t$ is Lipschitz continuous with respect to the trajectory-level MMD of the policy distributions. This seems to be a very strong assumption and could be potentially unrealistic in clinical settings where threshold-based dynamic policies might introduce discontinuities in physiological responses. In the paper the computational complexity of the MMD and MDS steps is not clear. Constructing the distance matrix would require pairwise MMD computations across all K policies at every time step t. If the set of policies is large, scaling could be poor. Could you please explain if you have looked into the choice of kernel bandwidth in the RKHS embedding (which is known to affect MMD calculations)?

2. Presentation
(1) The problem setup is explained well and the authors clearly articulate the distinction between single policy evaluation and multi-policy contrast.
(2) It is not clear how the theoretical bounds on the MDS-reduced vectors translate to the finite dimensional RNN hidden states. I found that separating the results and discussion of deterministic and dynamic policies makes the narrative fragmented.

3. Significance:
(1) The performance gains on the semi-synthetic datasets are demonstrated, whereas the real world case study is limited. While it’s no surprise that higher MAP targets lead to higher lactate, this finding doesn't really prove the model can handle complex and multi-dimensional treatment strategies beyond simple threshold tweaking.

4. Originality:
The Transformer backbone + G/Q heads relies heavily on the existing DeepLTMLE framework. The novelty lies in the preprocessing and the concatenation of the policy embedding. To rigorously evaluate the added value of the RKHS embedding, the authors could include a standard multi-task baseline (a shared backbone with K distinct Q-heads). This comparison is important for distinguishing the benefits of the specific embedding method from the inherent advantages of representation sharing.

---

> ### Author Rebuttal · Authors · 2026-03-31
>
> We thank the reviewer for raising these insightful points. We address them as follows:
>
> **Q1 (Lipschitz)**
> We clarify that Assumption B2 does not require policies themselves to be smooth (e.g., threshold rules can be discontinuous), but assumes the true Q-function varies smoothly with the policy-induced treatment distribution. This is a standard assumption in prior work [1–3]. Intuitively, policies inducing similar treatment distributions should yield similar counterfactual mean outcomes. For example, shifting a MAP threshold from 65 to 66 mmHg may affect boundary cases, but should not substantially change the population-level conditional mean outcome.
>
> [1] Marmarelis, Myrl G., et al. "Off-policy predictive control with causal sensitivity analysis." The 41st Conference on Uncertainty in Artificial Intelligence. 2025.
>
> [2] Khan, Samir, Martin Saveski, and Johan Ugander. "Off-policy Evaluation Beyond Overlap: Sharp Partial Identification Under Smoothness." International Conference on Machine Learning. PMLR, 2024.
>
> [3] Lecarpentier, Erwan, et al. "Lipschitz lifelong reinforcement learning." Proceedings of the AAAI Conference on Artificial Intelligence. Vol. 35. No. 9. 2021.
>
> **Q1 (Complex clinical care)**
> PEQ-Net supports general policies that map patient history to actions, including multi-covariate logic and temporal rules. While comprehensive clinical validation is important, it is beyond the scope of this work, which focuses on methodological advances for policy comparison. To address this concern, we add a real-world case study with a more complex discontinuation rule: stop if (MAP$>$m+10) or (MAP$>$m and SBP$>$s) for two consecutive windows. This goes beyond single-variable thresholding and shows that PEQ-Net can handle multi-covariate, temporally structured policies, and can be extended to more complex decision rules (results: <https://anonymous.4open.science/r/PEQ-Net-EA23/viz/capo_map_sbp_target.png>).
>
> **Q2 \& L2 (Scalability)**
> We refer the reviewer to our response to W6L6 (W1) for a detailed discussion on scalability w.r.t the number of policies.
>
> **Q3 (Kernel Bandwidth)**
> We find performance robust to bandwidth choices. At each time step, we randomly sample 500 history-action pairs and take the median of all pairwise Euclidean distances (denoted as 'med'). We then set the bandwidth to 1/(2med). For sensitivity analysis, we vary it as γ/(2med) with γ∈{0.01,0.1,1,10,100}. RMSE on the expanded DGP remains stable across settings:
>
> |γ|CF1b|CF2b|
> |--|--|--|
> |0.01|0.18|0.28|
> |0.1|0.16|0.28|
> |1|0.17|0.28|
> |10|0.15|0.29|
> |100|0.18|0.26|
>
> **Q4 (Multi-head Ablation)**
> We refer the reviewer to our response to U3sJ (W2) for this multi-head baseline.
>
> **Presentation**
> The theoretical guarantee applies to the parameterized outcome model using MDS-reduced vectors as input. The RNN module is introduced as a design choice to aggregate the sequence of MDS-reduced vectors, but the theory is not intended to characterize the RNN hidden states directly.
>
> We agree that the separation of deterministic vs. dynamic policies disrupts narratives. This separation is necessary since most baselines only support the former, making joint comparison unclear. To improve clarity, we will revise the manuscript as follows:
> (1) Remove the paragraph at lines 313-319 ("The policy encoder...policy embedding").
> (2) Rewrite the paragraph at lines 342-349 ("Following the...architectural changes") as:
> "Most baselines are designed for (a), while PEQ-Net accommodates both (a) and (b). Accordingly, we evaluate the two cases separately.
>
> For (a), we follow the setup of DeepACE, where all baselines are applicable. In this case, we bypass the MDS step and directly input the deterministic sequence into the policy encoder, treating it as a degenerate policy embedding.
>
> For (b), we restrict comparisons to DeepLTMLE, which is designed to handle dynamic policies without architectural changes.
> "
>
> **L1 \& L3 (Identifiability assumptions)**
> We will include the following discussion in the revision: "PEQ-Net relies on standard identifiability assumptions (e.g. no unmeasured confounding and positivity) in existing approaches based on g-formula (e.g. DeepACE, DeepLTMLE). In clinical settings, these assumptions may be violated (e.g., unrecorded factors or low overlap), leading to biased estimates. Our work focuses on improving multi-policy estimation under these assumptions and does not address their violation. Developing methods that are robust to their violation is an important direction for future work."
>
>
> **L4**
> We agree that deploying off-policy evaluation in ICU raises important risks. PEQ-Net is not intended to provide autonomous treatment recommendations. Rather, we view it as a tool for retrospective policy evaluation and clinical hypothesis generation, helping prioritize candidate strategies for further study. Careful human oversight and external validation are required before clinical use. We will clarify this point in the conclusion of our paper.

---

> > ### Author Rebuttal · Reviewer_f8ah · 2026-04-01
> >
> > Thank you very much for the rebuttal. The added information is helpful. It addresses some of my previous concerns. That said, I still have some reservations. My main one is the smoothness assumption behind the theorem. It still feels fairly strong for longitudinal clinical settings, where small policy changes may sometimes lead to non-smooth downstream effects. I also remain unclear on how fully the theoretical guarantee matches the implemented architecture, since the practical model includes an RNN aggregation step beyond the formal setup. The real-world case study is also still somewhat limited. For these reasons, I keep my original score.

---

> > > ### Author Response · Authors · 2026-04-03
> > >
> > > Thank you for acknowledging our rebuttal. We provide the following detailed clarification for your remaining concern:
> > >
> > > **Smoothness Assumption**
> > > Assumption B.2 concerns the stability of outcomes under small perturbations in **induced treatment trajectories**, not the smoothness of the policy rule itself. We will clarify this distinction in the revision.
> > >
> > > We note that the relationship proceeds in two steps:
> > > 1. change in policy rule → change in the policy-induced distribution of treatment trajectories
> > > 2. change in that distribution → change in the conditional mean outcome.
> > >
> > > Assumption B.2 does **not** require that small changes in the policy rule itself lead to small changes in outcomes (both steps). Rather, it assumes that small changes in the *policy-induced distribution of treatment trajectories* lead to small changes in the *mean outcome* (only the second step).
> > >
> > > For example, consider two threshold policies (e.g., MAP $>$ 100 vs. 101 mmHg). If this change only affects a small fraction of patients’ treatment trajectories, the induced treatment distributions are nearly identical. Under B.2,  the true conditional mean outcomes should also be similar.
> > >
> > > Assumption B.2 **does not** preclude the following possibilities:
> > > - a *small* change in policy rule (e.g., delaying treatment for many patients) can lead to *large* shifts in treatment trajectories
> > > - a *large* change in policy function may lead to *small* changes in treatment trajectories.
> > >
> > > These cases highlight that the assumption is defined over the induced treatment distribution, rather than the policy rule itself.
> > >
> > > A violation of this assumption would imply that changing treatment for a negligible fraction of patients leads to arbitrarily large shifts in the population-level outcome. For example, in vasopressor management, slightly adjusting a MAP threshold (e.g., 65 to 66 mmHg) typically only affects patients near the decision boundary, a negligible subset, yet the violation would imply that this subset leads to a substantial shift in the population-level outcome. Such behavior is generally unlikely in clinical settings, and the assumption can thus be viewed as a mild condition.
> > >
> > > **RNN aggregation.** We thank the reviewer for following up. We would like to clarify that the RNN is merely a design choice for aggregating per-step embeddings. Our theory **holds** for *any sequence model that respects the same tail-conditioning structure and satisfies Lipschitz continuity.*
> > >
> > > The theory does **not** require the sequence model to preserve trajectory-level distances. Instead, the guarantee follows from the relationship between **trajectory-level MMD** (used in the analysis) and **per-step MMD** (used in implementation). Lemma B.1 shows that per-step MMD is bounded by trajectory-level MMD up to a constant factor.
> > >
> > >
> > > Thus, if two policies induce similar trajectory-level treatment distributions, they also induce similar **per-step treatment distributions**, leading to similar per-step embeddings.
> > > Any sequence model—RNN, Transformer, etc.—that aggregates these embeddings will receive similar inputs, and by Lipschitz continuity, produce similar Q-functions.
> > >
> > > We use trajectory-level MMD in the theorem for analytical convenience: the second-order remainder involves policy differences across multiple sub-trajectories, and a single full-trajectory metric yields a cleaner bound. Directly computing trajectory-level MMD in practice would require estimating full counterfactual trajectory distributions, which is much harder than estimating the counterfactual mean outcome.
> > >
> > > Therefore, the RNN is not required by the theory. It is simply one convenient aggregator, and any sequence model respecting the tail-conditioning structure would preserve the guarantee.
> > >
> > > **Real-world Case Study.** We agree with the reviewer that the current real-world case study is not comprehensive. However, this is somewhat beyond the scope of an ML conference paper, as our goal here is to demonstrate that the proposed framework can be applied to realistic longitudinal settings and extended to policies of increasing complexity (as shown in our additional case study).
> > > More broadly, our method operates within the ICE G-comp framework, which reduces the problem to sequential prediction of conditional mean outcomes. This formulation allows the use of flexible models to process high-dimensional, multi-modal, time-varying clinical data. As such, the ability to handle complex clinical settings primarily depends on the expressiveness of the underlying model backbone, rather than the policy embedding itself.
> > >
> > > We leave a more comprehensive evaluation of our method in practical clinical settings as an important future direction, which we aim to target for a clinical journal.

---

### Official Review · Reviewer_U3sJ · 2026-03-13

**Soundness:** 3
**Presentation:** 4
**Significance:** 3
**Originality:** 3
**Overall Recommendation:** 4
**Confidence:** 4

**Summary:**

This paper proposes a new architecture for causal effect estimation under multiple policies in longitudinal settings. In particular, it introduces a policy-aware reparameterization of the iterative conditional expectation Q-functions, enabling shared representations across policies and thereby supporting more efficient estimation under multiple policies.

**Compliance With Llm Reviewing Policy:**

Affirmed.

**Final Justification:**

After reading through all the reviewer's comments and the authors’ responses, I have decided to maintain my original score and recommend a weak accept.

**Key Questions For Authors:**

I do not have any specific questions for the authors.

**Limitations:**

yes

**Strengths And Weaknesses:**

Strengths:

1. The paper is organized and written well so it is easy for readers to follow and understand.
2. It proposes a creative way to efficiently estimate the causal effect estimation in longitudinal settings, specifically advance the classical iterative conditional expectation method in multi-policy setting. The proposed method is clearly formulated and well motivated.
3. The paper provides sufficient detail on the data-generating process and method implementation, and the code is also available, which supports the reproducibility of the proposed method.

Weaknesses:

1. In Algorithm 2, the neural network parameter $\theta$ is updated after iterating over all time points and all policies. In the classical iterative conditional expectation, the Q-function at each time t is fitted separately, since the target Q-values at time t can only be obtained after fitting the Q-function at the next step and using its predictions under the policy as the regression target. In that setting, the target Q-values are fixed and interpretable. By contrast, under joint training across all time points, the target Q-values change as the parameter $\theta$ is updated during the optimization, making it unclear what the model is converging to and whether it converges to the true target Q-values. In addition, could joint training of all time points cause the optimization to converge to the same local minimum across time points, leading to overly similar or even nearly constant Q-values?
2. The policy embedding helps learn policy representations that support joint estimation across multiple policies. However, the paper lacks an ablation study to isolate the contribution of the kernel mean embedding component. It would be helpful to include visualizations of the learned embeddings, and the experimental evaluation would be strengthened by comparing effect estimation performance with and without the embedding.

Minor comments:

1. Typo in line 275, "Traning" to "Training"
2. If the kernel embedding is pretrained and kept fixed, the updated parameter $\theta$ in Algorithm 2 should not contain $\rho$ in the algorithm.

---

> ### Author Rebuttal · Authors · 2026-03-31
>
> **W1**: We thank the reviewer for raising a valid concern about the scheme of joint training across time steps. Our response is threefold:
>
> - Our model preserves the ICE G-comp structure. Although we use a shared model across time steps, the underlying definition/interpretation of Q-functions remains unchanged, which are recursively defined via next-step conditional expectations. As shown in Eq.(4) and Algorithm 2, the recursive dependency is preserved through the construction of regression targets, which still rely on the next-step Q-function. This shared parameterization improves data efficiency across time steps. In practice, such shared architectures consistently outperform separate per-step models and have become the dominant design in modern ICE-based methods, including DeepACE, DeepLTMLE, GST-UNet.
>
> - Moving target issue. We acknowledge that joint training introduces a moving-target issue, since the regression targets depend on the same parameters being optimized. To address this, we now introduce a lagged target network, a standard heuristic in deep reinforcement learning. In particular, we maintain a lagged copy of the parameters $\theta'$ and use it to compute the regression target $\hat{Q}_{t+1}$ for the regression loss, while using gradient descent to update parameter $\theta$. The copy is slowly updated via Polyak averaging ($\theta' = 0.005 \cdot \theta + 0.995 \cdot \theta'$) at the end of every batch. This technique decouples the target from immediate parameter updates, so each optimization step minimizes a fixed-target regression loss, analogous to stagewise ICE.
>
> - No collapse across time steps. A trivial collapse to a constant function is inconsistent with the training objective: At the final time step, the Q-function directly regresses on the observed outcome $Y$, which is non-constant, and a constant predictor would incur non-trivial loss. This signal propagates to earlier steps through the recursive targets, preventing collapse across time. Empirically, we also do not observe such behavior. As a diagnostic, we compute the Pearson correlations of Q-function outputs across time steps (<https://anonymous.4open.science/r/PEQ-Net-EA23/viz/Q_pearson_corr_heatmap.png>),
> The heatmap shows a clear structure: correlations are high for nearby time steps but decrease as the temporal gap increases. This indicates that the model captures temporal smoothness while still learning distinct Q-functions across time, rather than collapsing to a constant or nearly identical representation.
>
> **W2**: We thank the reviewer for this valuable suggestion.
> We first provide a visualization of the learned policy embeddings (<https://anonymous.4open.science/r/PEQ-Net-EA23/viz/policy_embedding_pca_by_time.png>). The plot shows a consistent geometric structure across time steps: policies with similar intervention patterns are mapped to nearby representations, while more dissimilar policies are clearly separated. This indicates that the embedding captures meaningful relationships between policies in a structured manner.
>
> To isolate the contribution of the policy embedding, a naive ablation that removes policy embedding would collapse all policies into a single task, making policy-specific $Q$-function estimation ill-posed. Instead, inspired by reviewer f8ah's suggestion, we construct a stronger and more appropriate baseline that preserves policy-specific outputs without using embeddings: a shared backbone with $K$ distinct Q-heads, one for each policy. This design maintains the ICE G-comp structure while modeling each policy with a separate head, without any explicit conditioning on policy representations. It allows us to isolate whether the performance gain arises from (i) parameter sharing alone or (ii) the embedding structure.
>
> To further demonstrate the effect of conditioning on policy-induced distributions (rather than using separate heads), we design a controlled experiment where policies differ only in early time steps: the three policies use thresholds $0.4$, $0.5$, and $0.6$ in the first two time steps, and share the same threshold ($0.5$) thereafter. Such time-varying policies are common in clinical practice, where treatment regimes are adjusted over time. We then evaluate the CATE between the $0.4$ vs $0.5$ (CF 1c) and $0.6$ vs $0.5$ (CF 2c) policies. Results over 20 runs show that PEQ-Net substantially reduces RMSE compared to the multi-head baseline:
>
> | Method     | Bias(CF 1c) | Bias(CF 2c) | RMSE(CF 1c) | RMSE(CF 2c) |
> |------------| ------------|-------------|-------------|-------------|
> |Multi Q-head|0.0409 ± 0.0328| 0.0280 ± 0.0206|0.0519  |0.0345  |
> | PEQ-Net    |0.0011 ± 0.0011| 0.0096 ± 0.0095|0.0015  |0.0133  |
>
>
> These results indicate that the improvement is not solely due to parameter sharing, but arises from leveraging structured similarities between policies via the embedding.
>
> **Typo**: We thank the reviewer for catching the typos. We will correct them in the final version.

---

> > ### Author Rebuttal · Reviewer_U3sJ · 2026-04-02
> >
> > Thank you for providing the rebuttal. It is helpful in clarifying and addressing my concerns. The additional experimental results are clear and provide sufficient support for the claim. I still have some concerns regarding the design of jointly training a single model for all Q-functions. Since the Q-function represents the conditional expectation of the outcome under intervention 𝑔, when the pseudo-outcomes are constructed and each Q-function is fitted separately, it is clear what target each model is converging to, and it is also easier to diagnose the performance of each model. In contrast, when all Q-functions are trained jointly, it is less clear whether the model converges to the correct solution. It is encouraging to see the use of a lagged target network to mitigate the moving-target issue. I encourage the authors to include additional experimental evidence on this aspect in a future revision. I kept my original score.

---

> > > ### Author Response · Authors · 2026-04-06
> > >
> > > Thank you for the clarification and for acknowledging our rebuttal. We agree that training separate Q-functions makes the target of each model easier to interpret and diagnose. For a jointly trained model, directly verifying the convergence of each learned Q-function is substantially more involved. Doing so would require approximating the true conditional mean outcome for a large collection of histories across multiple time steps, which in practice entails extensive nested Monte Carlo simulation from many intermediate steps. This is computationally expensive and beyond what we can complete during the rebuttal period. We therefore leave this as an important direction for future revision, and we appreciate the reviewer for highlighting it.

---

### Decision · Program_Chairs · 2026-04-30

**Decision:**

Accept (regular)

**Comment:**

The paper introduces a framework for longitudinal causal effect estimation when evaluating multiple dynamic treatment policies. This paper addresses a critical practical bottleneck in longitudinal causal inference—finite-sample variance inflation from separate estimation—by proposing a highly novel method (PEQ-Net) that creatively integrates RKHS-based policy embeddings into the ICE G-computation framework. The approach is theoretically sound, clearly formulated, and empirically demonstrates consistent advantages over baselines on both semi-synthetic MIMIC data and a real-world sepsis case study. Most of the reviewers' concerns were satisfactorily addressed in the rebuttal. However, the authors are encouraged to incorporate the reviewers' suggestions in the final version.